



# Retrieval of the Sea Spray Aerosol Mode from Submicron Particle Size Distributions and Supermicron Scattering during LASIC

Jeramy L. Dedrick[1], Georges Saliba[2], Abigail S. Williams[1], Lynn M. Russell[1], Dan Lubin[1]

[1]Scripps Institution of Oceanography, University of California, San Diego, La Jolla, California, USA
[2]Pacific Northwest National Laboratory, Richland, Washington, USA

*Correspondence to*: Lynn M. Russell (lmrussell@ucsd.edu)

**Abstract.** Improved quantification of sea spray aerosol concentration and size is important for determining aerosol effects on
clouds and climate, though accurately capturing the size distribution of the sea spray mode remains limited by the availability
of supermicron size distributions. In this paper we introduce a new approach to retrieve lognormal mode fit parameters for a
sea spray aerosol mode by combing submicron size distributions with supermicron scattering measurements using a Mie
inversion. Submicron size distributions were measured by an Ultra-High Sensitivity Aerosol Spectrometer (UHSAS), and
supermicron scattering was taken as the difference between < 10 µm and < 1 µm 3-wavelength integrating nephelometer
measurements (NEPH). This UHSAS-NEPH method was applied during background marine periods of the Department of
Energy Atmospheric Radiation Measurement Layered Atlantic Smoke Interactions with Clouds (LASIC) campaign on
Ascension Island (November 2016 – May 2017) when the contribution of sea spray aerosol was expected to represent a large
fraction of the aerosol mass and total scattering. Lognormal sea spray modal parameters were retrieved from comparisons
between nephelometer measurements and a look-up table of Mie theory-simulated scattering coefficients for low error
solutions that minimized the 0.4 – 1 µm residual in the UHSAS size distribution. The UHSAS-NEPH method retrieved sea
spray mode properties for approximately 95% of background periods during LASIC when scattering variability was low and
particle concentrations were typical of the clean marine boundary layer (< 400 cm⁻³), ranging from 0.6 to 1.6 µm in mass mean
diameter (1.3 ± 0.15 µm), 1.2 to 3.7 in modal width (2.2 ± 0.2), and mass concentration of 0.13 to 20.7 µg m⁻³ (6.6 ± 3.5 µg
m⁻³). The measured nephelometer scattering at 3 wavelengths was found to only marginally constrain the mode width at the
largest particle sizes in the absence of additional size and chemical measurements for defining parameters for the Mie solutions.
Comparing UHSAS-NEPH retrievals to those of a fitting algorithm applied only to the submicron UHSAS number size
distribution showed that correlations between retrieved mass concentration and the available mass-based sea spray tracers
(coarse scattering, wind speed, and chloride) are low when supermicron measurements are not considered. We also show that
measured supermicron size distributions are needed to adequately characterize the sea spray number concentration, though
mass concentration can be comparably characterized using the supermicron scattering. This work demonstrates the added value
of supermicron scattering measurements for retrieving reasonable sea spray mass concentrations, providing the best-available,





observationally-constrained estimate of the sea spray mode properties when supermicron size distribution measurements are not available.

## 1 Introduction

Sea spray aerosol contributes the largest natural source of particles to the global aerosol mass budget (Lewis and Schwartz, 2004). Wind-driven breaking waves and bubble bursting at the ocean surface produce sea spray particles composed of organic components and sea salts that are injected into the atmosphere (Odowd et al., 1997; Russell et al., 2010). Field measurements have shown that sea spray aerosol makes up 10 – 30% of the particles necessary for cloud formation, known as cloud condensation nuclei (CCN), at low supersaturations in marine regions (Modini et al., 2015; Quinn et al., 2017; Sanchez et al., 2021) and thus have important implications for modeled cloud properties and climate feedbacks (Horowitz et al., 2020). Model predictions of sea spray concentration are determined by a number of different emission parameterizations (e.g., (Gong, 2003; De Leeuw et al., 2011; Salter et al., 2015)), which leads to uncertainties in the sea spray mass production (2.2 – 118 $10^{12}$ kg yr$^{-1}$; (De Leeuw et al., 2011)), the shortwave scattering direct climate effect (-2.2 – -0.15 W m$^{-2}$; (Ayash et al., 2008)), and the aerosol-cloud indirect climate effect (-2.9 – +0.3 W m$^{-2}$; (Paulot et al., 2020)).

In order to reduce these uncertainties, measured sea spray concentration and size have been retrieved and correlated with environmental parameters such as wind speed and sea surface temperature for improved emission parameterization (e.g. (Saliba et al., 2019; Liu et al., 2021). The approach for this task is the combination of sizing instruments that resolve the sea spray aerosol size distribution across sub- and supermicron diameters, typically with a differential mobility analyzer (DMA) for 10 nm to 1 µm and an aerodynamic particle sizer for 0.5 to 10 µm diameter (APS; (Modini et al., 2015; Saliba et al., 2019; Quinn et al., 2017)). This technique has uncertainties controlled by the limited size range, resolution, and timing of each instrument, as well as by the ambient conditions. Merging mobility and aerodynamic measurements requires varying the particle density and shifting the size distribution until there is agreement between both instruments in the overlapping diameter range (Khlystov et al., 2004) due to uncertainties in particle densities of marine aerosol (Tang et al., 1997). The generally low number concentration of sea spray aerosol at supermicron sizes also causes poor counting statistics in the largest size bins of DMAs, which impacts the range of overlap to which the retrieval is sensitive (Russell et al., 1996a; Russell et al., 1996b). Although supermicron sea spray aerosol concentrations are low in number, their large mass contribution at this size have been shown to have non-negligible impacts on optical and cloud properties in the marine boundary layer (Dror et al., 2020; Murphy et al., 1998).

Given the uncertainties associated with multiple instrument size distribution merger and the limited availability of supermicron size distribution measurements in marine regions, alternative methods should be considered to adequately characterize the





modal properties of sea spray aerosol. The relative availability of nephelometer-based supermicron scattering measurements
provides an attractive alternative to supermicron size distribution measurements. The premise of the approach proposed here
is the observation that sea spray particle mass is often strongly correlated to the supermicron scattering during clean marine
conditions (Kleefeld et al., 2002; Chamaillard et al., 2006; Quinn et al., 1998). To translate scattering measurements to
equivalent size distributions, Mie theory (Mie, 1908) can be leveraged with three different scattering wavelengths to constrain
a single lognormal mode that is representative of the sea spray size distribution (Lewis and Schwartz, 2004), which has been
supported by a variety of ambient data sets fitting sea spray modes to measured size distributions (Modini et al., 2015; Saliba
et al., 2019; Quinn et al., 2017; Sanchez et al., 2021). This approach is a type of Inverse Mie Method (IMM) wherein
assumptions of particle size, composition, and concentration are made to determine optical properties of the particle population
(Bluvshtein et al., 2017). Similar approaches that combine observed or simulated size distributions with scattering
measurements have (Frie and Bahreini, 2021; Shen et al., 2019; Demott et al., 2016; Lv et al., 2018) found that optical
properties alone without the addition of a particle sizer are not always sufficient to estimate properties of the aerosol size
distribution.

In this work, we retrieve sea spray aerosol modal properties by fitting a single lognormal mode constrained by supermicron
scattering at 3 wavelengths to measured mass size distributions in the diameter range of 0.4 to 1 µm. This diameter range often
appears as a "shoulder" in measured number and mass size distributions and is largely composed of sea salt particles during
marine conditions (Quinn et al., 2017; Saliba et al., 2019; Zheng et al., 2018; Sanchez et al., 2021; Murphy et al., 1998).
Scattering-constrained lognormal modes that are fit to the measured size distribution are referenced from a look-up table of
Mie theory-simulated scattering coefficients using a combination of fitting parameters that define the concentration, mean size,
and width of the sea spray mode. Employing look-up tables in optical measurement inversions has been demonstrated in
previous inversion procedures (e.g. (Lv et al., 2018; Veselovskii et al., 2002)) and is based on minimizing error between
measurement and reference values for each retrieval to obtain a single solution. Recognizing the inherent restrictions of solely
minimizing error to derive a single, unique solution, namely the uncertainties in comparing theory estimates and measurements,
we incorporate known instrument error and measurement variability into the retrieval methodology. Within averaging
intervals, uncertainty and variability have been shown to impact the assumed size distributions and optical properties of aerosol
using inversion techniques (Viskari et al., 2012; Frie and Bahreini, 2021). This scheme enables a variety of solutions to be
aggregated and assessed so that a consistent and unique solution can be obtained by averaging solutions that are the most
statistically probable. Since sea spray aerosol concentrations are most relevant to CCN in "clean" marine environments (Quinn
et al., 2017) and the addition of non-marine sources (e.g. dust) tend to mask supermicron sea spray contributions, we directed
our method at measurements that are largely reflective of clean marine conditions at a location that lacked supermicron size
distribution measurements. This study first describes the measurements utilized for sea spray mode retrieval, followed by a
summary of the method employed to quantify modal properties from these measurements. Retrieved sea spray mode properties



are then compared with a submicron-only fitting algorithm, known sea spray tracers (supermicron scattering, wind speed, and chloride), and literature-reported values to evaluate the extent to which the retrieved mode is largely from sea spray.

## 2. Measurements

### 2.1 LASIC

Measurements from the Department of Energy Atmospheric Radiation Measurement (DOE ARM) site on Ascension Island, Saint Helena (7.96696° S, 14.34981° W) during the Layered Atlantic Smoke Interactions with Clouds (LASIC) campaign are used to demonstrate the performance of the sea spray mode retrieval. LASIC captured the annual and seasonal cycles of aerosol and cloud properties during an 18-month (April 2016 – October 2017) deployment of the ARM Mobile Facility 1 (AMF1) (Miller et al., 2016; Zuidema et al., 2016). AMF1 measurements were collected at an isolated site on the windward flank of Green Mountain, away from the island's airport and other inhabited areas (Zhang and Zuidema, 2019). The prevailing wind direction measured by meteorological instrumentation during the campaign was 115 ± 10° (east-southeasterly), indicating persistent sampling of offshore maritime air. The Mobile Aerosol Observing System (MAOS), a component of AMF1, housed the instruments, described in the following subsections, from an inlet situated 10 m above ground level at an altitude of 365 m above sea level (Uin et al., 2019).

Episodic intrusions of airborne biomass burning particles are carried into the Ascension Island marine boundary layer from South African wildfires annually during June – October (Zuidema et al., 2016). These events contrast sharply with the clean boundary layer that persists for the remainder of the year (November – May; (Pennypacker et al., 2020). Non-marine aerosol particles are limited during this "background" season, though a few transport events of African dust that entrain into the boundary layer have been documented as occurring during austral summer and fall months (January – April) in the Southeast Atlantic (Kishcha et al., 2015; Virkkula et al., 2006). For background (non-biomass burning) times without dust events, the aerosol population is expected to be largely from marine sources, of which sea spray represent a large fraction of the aerosol mass concentration. Analyzed measurements in this work will focus on LASIC background season observations (November 2016 – May 2017) (Table 1).

### 2.1.1 Submicron Particle Size Distributions

Two particle sizing instruments were operated during the LASIC campaign: a TSI Scanning Mobility Particle Sizer (SMPS; TSI Inc., Shoreview, MN, USA) and a DMT Ultra-High Sensitivity Aerosol Spectrometer (UHSAS; Droplet Measurement Techniques (DMT) Inc., Longmont, CO, USA). The SMPS measured the aerosol size distribution ranging from 10 nm to 460 nm dry mobility diameter, which did not include the submicron accumulation mode shoulder (0.4 – 1 µm) in mass size distributions (Fig. 1), meaning that it did not provide constraints on the sea spray mode retrieval and was not used here.



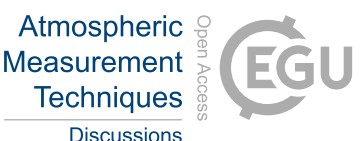

The UHSAS operated with 99 size channels at logarithmic spacing to cover diameters from 60 nm to 1 µm dry optical diameter
at a temporal resolution of 1 Hz that were averaged to 1 min. The UHSAS was calibrated using polystyrene latex spheres with
refractive index of approximately 1.59 and has a particle counting efficiency of approximately 100% for particle concentrations
below 3000 cm$^{-3}$ and sizes larger than 0.1 µm (Cai et al., 2008). The counts of particles per bin were converted to number size
distributions using the sample flow rate (typically 50 cm$^3$ min$^{-1}$) and the sample accumulation time (10 s).

UHSAS artifacts at large size bins have been reported for measurements in marine air masses (Pennypacker and Wood, 2017;
Sanchez et al., 2021). These artifacts appear as two consistent and narrow modes at dry optical particle diameters of 0.6 and
0.85 µm, which likely represent the splitting of the sea spray mode by partial drying of salt that has been sampled from high
ambient relative humidity (Fig. 2). These two modes constitute low contributions to the particle number concentration (Fig.
2a), but an appreciable amount to the mass concentration (Fig. 2b) of the measured size distributions. We expect that the 0.6
µm mode is the dried part of the salt mode, similar to distributions reported by (Sanchez et al., 2021), while the narrow 0.85
µm mode is the remainder of the salt mode that is only partially dried. A treatment for these artifacts to fit the sea spray mode
is described in Section 3.4.

### 2.1.2 Supermicron Scattering

1 min averaged dry scattering coefficients (b$_{sca}$) were measured by a TSI 3563 3-wavelength integrating nephelometer at red
(700 nm), green (550 nm), and blue (450 nm) light wavelengths over an angular integration range of 7° to 170° (Anderson et
al., 1996) and impactor size cuts of 1 and 10 µm that alternated at intervals of about 1 h. Scattering coefficients were corrected
to account for known truncation errors due to significant coarse sea salt particle forward scattering at angles less than 7°
(Anderson and Ogren, 1998). The detection limit of the nephelometer for typical operating conditions is between 0.1 and 0.3
Mm$^{-1}$ depending on the wavelength (Anderson et al., 1996). Measured particles were heated and dried to relative humidity
(RH) below the average ambient value (ambient: 88 ± 8% RH; nephelometer internal: 60 ± 4% RH) upstream before scattering
was measured. Though these particles were not effectively dried to below the efflorescence point for sea salt mixtures (Ming
and Russell, 2001), they were similar to the humidity range of the dried UHSAS size distributions (55 ± 8 % RH). We chose
not to baseline size distributions or scattering coefficients to the same RH as the uncertainty of the RH values (~10%) likely
falls within the mean and range of reported humidity for each instrument. In addition, assumptions of particle composition
without chemical measurements to correct scattering coefficients would add further uncertainties to the methodology. 2-hr
averages of the scattering measurements for each impactor size cut were used to derive supermicron scattering coefficients at
each wavelength (b$_{sca,1-10µm}$(λ)) by differencing the b$_{sca}$ at 1 µm from b$_{sca}$ at 10 µm. Supermicron scattering during the LASIC
background season had an average value of 12.0 ± 6.3 Mm$^{-1}$ (0.3 – 41.1 Mm$^{-1}$) as measured by the nephelometer at 550 nm
and made up 70 ± 7% of the total scattering for particles less than 10 µm diameter. The combined use of the submicron particle



size distribution from UHSAS and the supermicron scattering coefficients from the nephelometer merit naming this method
UHSAS-NEPH.

### 2.1.3 Uncertainty and Variability of the Size Distributions and Scattering Measurements

Measurement variability and instrument error are incorporated into the sea spray mode retrieval to account for uncertainties in
the Mie theory-based inversion of scattering and size distribution measurements (Section 3).


UHSAS sizing uncertainty is within 2.5% of the particle size (Dmt, 2017) with variations of -10% to +4% based on calibrated
particles with known refractive index between 1.44 and 1.58 (Moore et al., 2021). The reported systematic uncertainty of
number size concentration for accumulation mode (0.1 – 1 µm) particles measured by UHSAS has been shown to be 3.9% due
to calibration, flow, and pressure biases (Kupc et al., 2018). This instrument error propagates to -27.5% to +12.4% for higher
moments of the size distribution such as surface area and volume (Kupc et al., 2018; Brock et al., 2019). We therefore adopt a
size uncertainty value ($\sigma_D$) of 2.5% as defined by the UHSAS instrument manufacturer (Dmt, 2017) and 10% for the
concentration uncertainty ($\sigma_{PNSD}$) that has been used in previous inversion procedures (Bluvshtein et al., 2017; Frie and
Bahreini, 2021). Measured size distribution variability was calculated for the UHSAS size distribution at each diameter bin
($\sigma_{PNSD,meas}(D_p)$) as the standard deviation of the 2-hr averages.


Systematic uncertainties of the particle scattering are mainly due to non-idealities at each measurement wavelength and angular
sensitivities of the nephelometer (Anderson and Ogren, 1998). These features promote the use of a scattering uncertainty
($\sigma_{sca,inst}(\lambda)$) value of 5% that has been used in previous inversion procedures (Frie and Bahreini, 2021; Bluvshtein et al., 2017).
Measured scattering variability was calculated for the supermicron scattering at each wavelength ($\sigma_{sca,1-10\mu m}(\lambda)$) as the standard
deviation during the 2-hr average. A list of the measured size distribution and scattering variables and their associated
uncertainties and variabilities are provided in Table 2. Particle losses due to aspiration and transmission in the MAOS were
assessed using the particle loss calculator (PLC (Von Der Weiden et al., 2009)), sample line configurations and geometry from
(Bullard et al., 2017), and a particle density of 1 g cm$^{-3}$. Losses were found to be less than 10% for particles smaller than 1 µm
diameter and greater than 50% for particles larger than approximately 6 µm. As a result of these losses we do not reconstruct
UHSAS size distributions, but note that nephelometer supermicron scattering measurements may underestimate retrieved sea
spray number and mass concentrations at the largest diameters. Using the mean sea spray mode statistics of UHSAS-NEPH,
this loss can equate to an underestimation of roughly $13 \pm 8\%$ in the sea spray mass and $0.8 \pm 4\%$ in number using nephelometer
scattering.





### 2.1.4 Ancillary Variables

As an alternative to size-resolved filter measurements of sodium, which were not collected during LASIC, measurements from an aerosol chemical speciation monitor (ACSM; Aerodyne Research, Billerica, MA USA) were used to evaluate sea salt mass retrievals. ACSM provided mass and chemical composition (organics, sulfate, nitrate, ammonium, and chloride) of non-refractory submicron aerosols. Since sea salt does not volatilize efficiently at 600°C, the ACSM measurement of non-refractory chloride provides a trace signal from NaCl in the absence of large sources of non-refractory chloride (Frossard et al., 2014;

Ovadnevaite et al., 2012) and has been used as a tracer to identify sea salt aerosol contributions to CCN (Humphries et al., 2021). We use chloride measurements only for January 2017 – May 2017, as these data were processed and quality assured at the time of evaluation. 2-hr average ACSM chloride concentration showed statistically significant positive correlation with two common sea spray tracers: wind speed (R = 0.21) and nephelometer supermicron scattering at 550 nm (R = 0.42) (Fig. S1). This observation supports the potential of LASIC ACSM chloride measurements to serve as a chemical tracer for sea

spray mass in the evaluation of our proposed retrieval method.

We additionally incorporated measurements of 1 min averaged condensation nuclei concentrations above 3 nm ($CN_3$) from a TSI Ultrafine Condensation Particle Counter (CPC) 3776, wind speed and rain intensity from a Vaisala WXT-520, refractory black carbon (rBC) concentration from a DMT Single Particle Soot Photometer, 470 nm particle absorption from a Radiance

Research Particle Soot Absorption Photometer (PSAP) and carbon monoxide (CO) mixing ratio from a Los Gatos Research trace gas analyzer (Miller et al., 2016). These measurements were used to identify clean marine periods (Section 2.2) and to assess environmental influence on retrieval performance. Ancillary measurements were averaged to 2-hr resolution to match the timing of the supermicron scattering coefficients and size distribution averages.

### 2.2 Clean Marine Periods

Non-marine aerosol particles, specifically those from combustion sources, have been shown to influence the performance of sea spray mode retrievals by contributing number and mass concentrations that overlap with the sea spray mode fitting region (Modini et al., 2015; Saliba et al., 2019). Here we focus on seasons when boundary layer air masses are assumed to have a multi-day marine history in order to reduce non-marine sources and ensure retrieval results that are consistent with sea spray. In the case of LASIC observations, boundary layer intrusions of biomass burning or dust aerosol can affect particle optical

properties by increasing absorption and reducing scattering for sub- and supermicron particles (Delene and Ogren, 2002; Denjean et al., 2020). In particular, UHSAS instruments show under-sizing of particles when highly absorbing biomass burning aerosol are introduced into the particle population due to heating from the instrument beam and subsequent particle vaporization and shrinkage (Howell et al., 2021). Although this effect is more likely to impact particles smaller than the assumed sea spray size ($D_p > 0.4$ µm), any impact on potential submicron sea spray aerosol contributions would have an

influence on our retrievals. Additionally, dust and other continental aerosol can be advected from southern central Africa and





into the Southeast Atlantic marine boundary layer and affect the size overlap of the fitting region used in the UHSAS-NEPH algorithm (0.4 – 1 µm) due to increased contributions of non-sea salt coarse mode particles in the size distribution (Miller et al., 2021). To limit these influences on UHSAS-NEPH retrieval, we isolated measurements during "clean marine" periods of the LASIC background season from November 2016 through May 2017 by applying the following criteria:


(1)  $CN_3$ concentration less than 600 cm$^{-3}$, which was the approximate 90th percentile particle concentration during the LASIC campaign,

(2)  CO mixing ratio (a proxy for continentally-sourced air) below the limit of ambient marine boundary layer background levels observed during LASIC (0.07 ppb; (Pennypacker et al., 2020)),


(3)  rBC concentration below the combustion source threshold of 50 ng m$^{-3}$ used by (Saliba et al., 2020) in the remote marine North Atlantic, and

(4)  < 10 µm Scattering Angstrom Exponent ($SAE_{10}$) values less than 1 for 450 and 700 nm nephelometer scattering. $SAE_{10}$ characterizes the wavelength dependence of particles and takes on small (< 1) values during periods in which coarse aerosol, such as sea salt, have a large mass contribution (Shen et al., 2019; Mulcahy et al., 2009).


We additionally removed periods when the rain intensity exceeded 1 mm h$^{-1}$ at any hour during the 2-hr average to ensure minimal influence of precipitation on the retrieval, namely wet scavenging of sea spray aerosol by rain droplets. 14 periods exceeded this rain intensity restriction. The combination of all criteria identified 1240 2-hr (non-raining) clean marine periods, which accounted for 50% of all available background season observations. The clean marine criteria reduced the average

values by 12% for $CN_3$, 7% for CO, 70% for rBC, and 15% for $SAE_{10}$ after applying these restrictions (Fig. 3). The 1240 available periods provided persistent marine conditions that were low in aerosol concentration (300 ± 90 cm$^{-3}$) and in combustion influence (15.7 ± 12.7 ng m$^{-3}$), as well as having a large scattering contribution from coarse particles ($SAE_{10}$ = 0.66 ± 0.15).

**3 Sea Spray Mode Retrieval (UHSAS-NEPH)**

This section outlines the procedure used to retrieve sea spray size distributions from scattering measurements and submicron mass size (Fig. 4). We describe the relationship of particle scattering to particle size (Section 3.1) and how this theoretical relationship is used to identify a group of probable sea spray mode solutions that are consistent with the measured supermicron scattering, as well as with literature reported ranges of modal properties (Section 3.2). Mode solutions are then constrained with measured submicron (0.4 – 1 µm) mass size distributions to isolate and retrieve the most consistent sea spray modal

properties (Sections 3.3 – 3.4). Using an external dataset that had supermicron size distribution measurements to evaluate our methodology, we show that the proposed scattering-constrained approach is comparable to the commonly used merger of size distribution measurements to fit a sea spray size distribution (Text S2). Differences in the modal parameter results were observed based on the differences in constraint (number-size versus mass-size and scattering), though both retrievals provide





results that are consistent with sea spray mode characterization and production. The MATLAB (The Mathworks Inc., version
R2019b) code to execute the sea spray retrieval outlined in this section can be downloaded from the UCSD digital archives
(https://doi.org/10.6075/J0GT5NCR).

## 3.1 Simulating Sea Spray Mode Scattering using Mie Theory

A look-up table of scattering coefficients was developed by employing a modified Mie theory code based on the algorithm for
red (700 nm), green (550 nm), and blue (450 nm) wavelengths (Bohren and Huffman, 1998). These wavelengths were chosen
to match those used by the 3-wavelength integrating nephelometer operated during LASIC (Section 2.1.2). Each of the three
red, green, blue (RGB) scattering coefficients is attributed to a combination of lognormal mode fitting parameters ($N_t$, $D_g$, and
$\sigma_g$) that describe the shape of the sea spray mode. We use the canonical lognormal mode form to represent the number size
distribution of sea spray aerosol with the following Eq. (1):

$$\frac{dN}{dlogD_p} = \frac{N_t}{\sqrt{2\pi}\,\log_{10}\sigma_g}\,e^{-\frac{(\log_{10}D_p - \log_{10}D_g)^2}{2(\log_{10}\sigma_g)^2}} \tag{1},$$

where $N_t$ is the number concentration of particles (cm$^{-3}$), $D_p$ is the particle diameter (μm), $D_g$ is the geometric mean diameter
(μm), and $\sigma_g$ is the geometric standard deviation (or mode width; unitless). These values are 1 – 99 cm$^{-3}$ for number
concentration, 0.05 – 1.19 μm for mean diameter, and 1 – 4 for mode width which provides over 157,000 possible sea pray
mode solutions (Table 3). The range of fitting parameters were chosen to reflect those reported in laboratory experiments and
field measurements (Table 3).

The scattering coefficients ($b_{sca,MIE}(\lambda)$) are then related to the size distribution by integrating Eq. (2) over all particle diameters
(0.01 – 10 μm),


$$b_{sca,MIE}(\lambda) = \int_0^\infty \frac{\pi D_p^2}{4}\, Q_{sca}(\lambda, m, D_p)\, \frac{dN}{dlogD_p}\, dlogD_p \tag{2}.$$

$Q_{sca}$ is the scattering efficiency, $\lambda$ is the light wavelength, and $m$ is the particle core refractive index ($m = n + ik$). Within the
Mie simulation, the refractive index, $m$, is assumed to be constant at $1.56 + 0i$, which is typical for sea salt particles (Bi et al.,
2018; Kent et al., 1983). This value provides a reasonable estimate of the scattering at marine sites where sea salts contribute
a majority of supermicron scattering mass, with organic components contributing minor if not offsetting differences to the
refractive index.



## 3.2 High Probability Mie Solutions

Since many solutions are within the constraints of three wavelengths of measurements, we use the error between the
nephelometer scattering and the Mie theory solutions to remove mode solutions that are not within the calculated error. The
probability of solutions that meet the error threshold are then evaluated and only the top 5% most probable are selected.

The measured supermicron scattering coefficient ($b_{sca,1-10\mu m}(\lambda)$) is compared to scattering coefficients computed for each
simulated sea spray size distribution and Mie theory ($b_{sca,MIE}(\lambda)$) in the look-up table by calculating the absolute error at each
wavelength ($\lambda$), using Eq. (3):

$$\Delta b_{sca}(\lambda) = \left| b_{sca,1-10\mu m}(\lambda) - b_{sca,MIE}(\lambda) \right| \tag{3}.$$

The scattering error ($\Delta b_{sca,RGB}$) is then computed by propagating the absolute error at each wavelength following Eq. (4):


$$\Delta b_{sca,RGB} = \sqrt{\sum_{\lambda=R,G,B}[\Delta b_{sca}(\lambda)]^2} \tag{4}.$$

The total sample space is then reduced by selecting solutions from the look-up table that fall below the error threshold ($\sigma_{sca,RGB}$)
calculated for that measurement time,


$$\Delta b_{sca,RGB} < \Delta \sigma_{sca,RGB} \tag{5a},$$

where,

$$\Delta \sigma_{sca,RGB} = \sqrt{\sum_{\lambda=R,G,B}\left[\left(\sigma_{sca,1-10\mu m}(\lambda)\right)^2 + \left(\sigma_{sca,inst}\right)^2\right]} \tag{5b}.$$

This error threshold incorporates the measured scattering variability at each wavelength for the averaging period ($\sigma_{sca,1-10\mu m}(\lambda)$)
and accounts for the instrument error ($\sigma_{sca,inst}$, which is constant). Figure 5 illustrates a time series of these error thresholds and
the percent reduction of the Mie solution sample space (N = 157,850) for each retrieval during the background season of the
LASIC campaign. Using this error threshold, the solution space is reduced by 98% on average with a range of 83 – 99%
reduction, resulting in approximately 1000 – 2000 possible solutions each time.

The majority of solutions that are below the error threshold typically have a sea spray mode shape similar to those previously
reported in literature, namely mass mean diameters within or near the coarse mode size range (1 – 10 µm) and mode width of



2 – 3, decreasing to below 0.1 cm⁻³ number concentration before the 10 µm cutoff. However, a few solutions meet the error threshold criterion, but are either too wide or large relative to the reported range of sea spray modes (Table 5, Fig. S2), which is a limitation of having only three scattering wavelengths to constrain the mode. To remove the outlier solutions and to restrict the sample space to a more consistent group of solutions, we sample only from sea spray modes with fitting parameters in the top 5ᵗʰ percentile of $N_T|\sigma_g$ joint probability (Text S1). The joint probability of $D_g|\sigma_g$ was also considered as a metric, but the

majority of retrieved modal properties using this combination consistently had mass mean diameters that were submicron in size ($< 0.6$ µm) and mode widths that were exceptionally broad ($> 3.4$), both of which are uncharacteristic of observed sea spray modes (Modini et al., 2015; Saliba et al., 2019; Quinn et al., 2017). The resulting most probable solutions from the $N_T|\sigma_g$ joint probability (typically $100 – 300$ for each measurement time) are then compared to the diameter range of $0.4 – 1$ µm of the UHSAS mass size distribution (Section 3.4).

**3.3 Perturbing the Size Distribution**

In order to account for the variability from averaging the measured size distribution over some time period when fitting the Mie solutions to the size distribution, random noise is introduced into the size distribution based on the size ($\sigma_D$) and number concentration ($\sigma_{PNSD,meas}(D_p)$, $\sigma_{PNSD,inst}$) uncertainties (Section 2.1.3).

Perturbations are simultaneously made to the measured number size distribution and particle diameters. This method uses the MATLAB function *normrn* (The Mathworks Inc., version R2019b) to introduce Gaussian noise into each bin of the size distribution by generating a random number sampled from a normal distribution. The sampled normal distribution is defined by mean (µ) and standard deviation (σ) parameters where µ is the measured PNSD averaged over the time interval ($\sigma_{PNSD,meas}(D_p)$) and σ is the combined errors of the temporal variability and instrument concentration uncertainty (which is

constant),

$$\sigma = \sqrt{\left(\sigma_{PNSD}(D_p)\right)^2 + (\sigma_{PNSD\ inst.})^2} \quad (6).$$

The diameters in the size distributions are perturbed by shifting the size bins by the same value using *normrnd* with a µ of 0

and σ as the instrument size uncertainty ($\sigma_D$). Size distributions are perturbed 100 times to provide a sample space of $N_{perturb} + 1$ (the measured size distribution and 100 perturbations). Each probable Mie solution retrieved in Section 3.2 is then tested for the $N_{perturb} + 1$ cases within the fitting region of the measured submicron size distribution described in Section 3.4.



### 3.4 Fitting Modes to the Measured Size Distribution

The last step in retrieving sea spray modal properties is to select a Mie solution that most closely matches the shape of the
measured size distribution in a specified fitting region. Considering the often strongly correlated relationship between
supermicron scattering and sea spray mass during clean marine conditions, all PNSDs are converted to particle mass size
distributions (PMSDs) using

$$\frac{dM}{dlogD_p} = \frac{\pi}{6} \rho D_p^3 \frac{dN}{dlogD_p} \tag{7},$$

with the assumptions of spherical particle homogeneity and unit density ($\rho = 1$ g cm$^{-3}$).

To account for the consistent UHSAS artifacts at 0.6 and 0.85 µm (Section 2.1.1), we modified the expected sea salt 0.4 – 1
µm diameter range used to fit the Mie theory-simulated size distributions to 0.38 – 0.83 µm (the closest UHSAS diameter size
bins within the specified range), which weights the comparison toward smaller sizes and effectively reduces the influence of
the largest artifact while maintaining the shape of the accumulation mode "shoulder". For each Mie solution (PMSD$_{MIE}$), a
residual sum of squares is computed between the measured and perturbed PMSDs as,

$$Fit\ RSS(j,k) = \sum[PMSD(D,j) - PMSD_{MIE}(D,k)]^2 \tag{8},$$

where index $j$ represents each measured or perturbed size distribution and $k$ is the index of the low error Mie solution. The fit
RSS provides a quantifiable metric for comparing the scattering-retrieved mode to the expected sea spray fitting region. Chi-
square "goodness of fit" calculations were also calculated and provided similar retrieved sea spray modal properties to those
retrieved using the fit RSS minimization. The low error-restricted Mie solution with minimum fit RSS for the measured and
noise-perturbed size distributions is chosen to establish a range of mode fits. This Monte Carlo approach allowed for the
selection of Mie solutions that capture the shape of the accumulation mode "shoulder" given uncertainties associated with
measurement variability and instrument error and provides a statistically robust sample space for retrieving a unique sea spray
mode solution. From the range of fit RSS-minimized sea spray modes, 95% confidence intervals of each fitting parameter are
calculated to further constrain the most probable solution that fits the measured size distribution in the expected sea spray size
range. On average, 30 – 40 solutions remain from this perturbation analysis for each measurement time with variabilities of
4% in number concentration, 3% in geometric mean diameter, and 1% in geometric standard deviation based on the sample
means and upper and lower limits of the 95% confidence intervals. The low variabilities of these fitting parameters demonstrate
consistent mode retrievals within the perturbations and stability of the retrieval procedure. Lastly, the fitting parameter
solutions that are both highly probable (Section 3.2) and within the 95% confidence interval of fit RSS-minimized modes are



averaged to produce a single lognormal sea spray size distribution with uncertainty of the solution defined as the upper and
lower bounds of the 95% confidence interval.

## 4 Performance of UHSAS-NEPH Retrievals

Sea spray modal properties were retrieved for 1237 of the 1240 2-hr background periods during LASIC using this algorithm
to simultaneously fit size distributions constrained by UHSAS and NEPH measurements (UHSAS-NEPH). The 3 periods for

which the sea spray mode could not be retrieved were due to missing supermicron scattering measurements for at least one
wavelength. In order to ensure that the algorithm retrievals were sufficiently consistent with both UHSAS and NEPH and
representative of marine aerosol, the results were restricted to those with (1) low residual errors between the retrieval and
measurements in the fitting region (0.38 – 0.83 µm), (2) low measured scattering variability during each 2-hr time period, (3)
dry or near-dry nephelometer relative humidity ($< 60\%$), and (4) limited influence from polluted sources, namely $CN_3$

concentrations $< 400$ cm$^{-3}$.

The fit RSS (Section 3.4) is the difference between the measured size distribution fitting region and the Mie solution
determined from 3-wavelength supermicron scattering. Low residuals (fit RSS $< 2$) indicate good agreement between the
measured region of the accumulation mode "shoulder" (0.38-0.9 µm) and the modal fit retrieved from scattering, hence low

RSS shows the retrieved mode is well constrained by the measured UHSAS size distribution and NEPH scattering. The average
fit RSS for the LASIC dataset that met the marine criteria was $2.4 \pm 1.4$, which indicates that the retrieved modes are generally
within the uncertainty of the measured size distribution in the 0.38 – 0.83 µm fitting region, but solutions that are not
constrained by the UHSAS size distribution are also retrieved as evident by large values of fit RSS for various retrieval periods
(Fig. 6a). A visual inspection of the retrieved mode fits suggests that values exceeding a residual threshold of 5 should be

rejected as there is not sufficient agreement in the overlap region to consider the solution acceptable. Retrieved modes with
fits above this threshold tended to have solutions that were either larger than the measured accumulation mode "shoulder" or
had peaks in regions that the size distribution was low. These high fit RSS likely indicate that the supermicron scattering
measurements were influenced by particles other than sea spray, which were not effectively constrained by the supermicron
scattering. In selecting the RSS threshold of 5, we also examined RSS thresholds of 1 through 10. There were no significant

changes in correlations to wind speed or chemical signatures (Section 5.2) for lowering the threshold to values of 1 − 4, and
there were decreases in those correlations for thresholds above a value of 6.

During the time period of March 27 – April 4 2017, many retrievals had fit RSS values above 5, possibly because they may
have been influenced by dust or other non-marine aerosol intrusions that were not effectively screened using the clean marine

criteria. To further ensure that this period was consistent with marine-source aerosol, we estimated the sub-10 µm single-
scattering albedo (SSA) from NEPH scattering and PSAP absorption. Sea salt aerosol typically have a high SSA (~0.99), while
the higher absorbance properties of dust and biomass burning decrease SSA to lower values (<0.95) (Muller et al., 2011). Due



to the predominantly course sizes of sea salt and dust, both are expected to have low scattering angstrom exponent values (SAE < 1) (Delene and Ogren, 2002). Distinguishing between the different aerosol types requires finding regimes in which both the

SSA is high (near 1) and the SAE is low (< 1). To estimate sub-10 µm SSA, NEPH scattering was adjusted to match the PSAP wavelength of 470 nm using the sub-10 µm scattering Angstrom exponent at 450-550 nm ($SAE_{10,450-550}$) (De Faria et al., 2021). The average $SAE_{10,450-550}$ and 470 nm SSA for this time period were $0.46 \pm 0.1$ and $0.98 \pm 0.01$ respectively, which align with the clean marine background season averages of $0.66 \pm 0.15$ ($SAE_{10,450-550}$) and $0.97 \pm 0.01$ (SSA). Additionally, we assessed if removing the observations during this period when the SSA was less than 0.95 would improve the sea spray tracer

correlations (Section 5.2). The SSA value of 0.95 was selected as threshold to distinguish dust influence following a previous characterization at the 470 nm wavelength (Muller et al., 2011). We found that removing these periods resulted in a marginal reduction in sea spray mass correlations with wind speed and scattering. Since this decrease in correlation and high average single-scattering albedo values indicate the retrieved modes were more consistent with sea spray than dust (Muller et al., 2011), no observations during this period.


To assess the limitations of the algorithm due to the observed variability in measured scattering, we examined the relationship between the scattering error threshold and the fit RSS. The scattering error threshold, $\Delta\sigma_{sca,RGB}$, is defined as the combined effect of temporal variability and instrument uncertainty and had an average value of $3.1 \pm 2.1$ Mm$^{-1}$ during LASIC. Low fit RSS (< 5) appear to coincide with low values of $\Delta\sigma_{sca,RGB}$, which were well-constrained sea spray mode solutions when the

nephelometer scattering variability was low during the 2-hr average (Fig. 6b). The relationship of low $\Delta\sigma_{sca,RGB}$ with low fit RSS generally persists to a threshold value of about $\Delta\sigma_{sca,RGB} = 5$ Mm$^{-1}$ before the fit RSS values increase in magnitude. Using this observed relationship between $\Delta\sigma_{sca,RGB}$ and fit RSS as a measure of the algorithm being sufficiently constrained to provide reasonable fits, we applied a restriction of acceptable scattering uncertainty tolerance of 5 Mm$^{-1}$.

The UHSAS and nephelometer instruments operated during LASIC sampled from a line that dried particles below ambient values before size and scattering were measured. For the combined UHSAS-NEPH retrieval, it is ideal if the NEPH relative humidity is as dry as the UHSAS ($55 \pm 8$ %). The average ambient relative humidity was $88 \pm 8$%, which was dried by the nephelometer to $60 \pm 4$%. This internal nephelometer humidity is above the level of sodium chloride efflorescence (45%) and below the level of deliquescence (75%; (Zhang, 2018). At higher relative humidity, supermicron sea salt particles could

alternate between efflorescence and deliquescence sizes, changing the supermicron scattering properties as they take up or evaporate water (Tang and Munkelwitz, 1994; Chamaillard et al., 2006). Transitions between these states during the 2-hr average as well as some incomplete drying of the particles from high ambient relative humidity could contribute to biases in the UHSAS-measured versus Mie theory-derived size distributions and scattering, and consequently to increases in fit RSS. Mode solutions with fit RSS above the specified value of 5 had average nephelometer relative humidity of $65 \pm 3$% (Fig. 6c).

Increased nephelometer relative humidity was associated with an increase in the fit RSS given by a statistically significant



(Student's t-test p-value < 0.05) positive correlation coefficient of 0.28 (Fig. 6c). Comparing the nephelometer scattering to the nephelometer relative humidity showed that supermicron scattering was independent of the instrument relative humidity up to a value of approximately 60% where the supermicron scattering coefficient begins to increase rapidly (Fig. S5). Because supermicron chemical composition measurements were not available, it was not possible to apply correction factors to correct

scattering values for particle humidification effects. Consequently, we restricted the retrievals to periods when the nephelometer relative humidity was below 60%.

Figure 6d illustrates a moderate correlation between fit RSS and $CN_3$ concentration (R = 0.5). Fit RSS is generally below the threshold of 5 for $CN_3$ concentrations less than 400 cm$^{-3}$. Above this concentration the fit RSS increases to higher values on

average. The 600 cm$^{-3}$ $CN_3$ criteria used to screen for clean periods is above the average for the clean marine background season (300 ± 90 cm$^{-3}$). Because supermicron sea spray particles contribute low number concentrations to the aerosol budget, increases in particle number concentration likely indicate non-marine pollution sources that were not excluded by the < 600 cm$^{-3}$ clean marine criteria applied here. Therefore we have excluded periods when the total aerosol concentrations exceeds 400 cm$^{-3}$ from the retrieval evaluation.


This additional screening of the LASIC dataset using $CN_3$ (< 400 cm$^{-3}$) and nephelometer relative humidity (< 60%) criteria provided 971 available 2-hr periods. Sea spray modal properties were retrieved for 95% (921) of these periods after applying the restriction of fit RSS (< 5) and scattering error tolerance (< 5 Mm$^{-1}$).

**5 Evaluation of UHSAS-NEPH Sea Spray Retrieval during LASIC**

Since sea salt composition measurements were not collected during the LASIC campaign, four methods were used to evaluate sea salt identification: (1) comparison of UHSAS-NEPH to a modified version of a sea spray size distribution fitting algorithm that has been validated previously with salt composition (Saliba et al., 2019; Modini et al., 2015; Quinn et al., 2017), (2) correlation of supermicron scattering and sea spray mass, where coarse scattering is taken as a tracer for sea spray during clean marine conditions (Kleefeld et al., 2002; Chamaillard et al., 2006; Quinn et al., 1998), (3) correlation of retrieved mass to wind

speed, since it is widely used as a proxy for sea spray mass production (Lewis and Schwartz, 2004) and model flux parameterization (Gong, 2003; De Leeuw et al., 2011; Ma et al., 2008), and (4) correlation to the non-refractory chloride signal measured by the aerosol chemical speciation monitor (ACSM).

**5.1 UHSAS-only Comparison**

We applied the fitting algorithm described above to UHSAS number size distributions (hereafter identified as UHSAS-only)

and compared sea spray mode results with those retrieved using UHSAS-NEPH. (Saliba et al., 2019) optimized this method for merged number size distributions from a differential mobility analyzer and aerodynamic particle sizer covering a dry particle size range of 0.02 – 10 µm in the remote North Atlantic. (Sanchez et al., 2021) have recently applied the algorithm to





submicron UHSAS size distributions obtained from aircraft measurements in the marine boundary layer of the Southern Ocean and found it to be a good approximation of sea spray contribution to CCN number concentration by comparison to quantified

sub- and supermicron sea salt particles using electron microscopy. The mode fitting parameters $N_T$ and $D_g$ of the UHSAS-only method were converted to mass-derived values for comparison with UHSAS-NEPH using Eq. (7) for the same particle diameter range ($D_p = 0.01 - 10$ µm). UHSAS-only sea spray modes were fit for 90% of the 2-hr average size distributions during the clean marine background season. Times that could not be fit generally included noise in the measured size distribution or other common singularities (see supplement of (Saliba et al., 2019)). Summary statistics comparing parameters

retrieved from these methodologies are provided in Table 4.

Integrated sea spray mode mass concentrations ranged from 0.008 to 42 µg m$^{-3}$ employing UHSAS-only and 0.13 to 20.7 µg m$^{-3}$ with UHSAS-NEPH. Average sea spray mass concentrations of $1.3 \pm 2.2$ µg m$^{-3}$ and $6.6 \pm 3.5$ µg m$^{-3}$ were observed for UHSAS-only and UHSAS-NEPH, respectively, which indicates that using UHSAS-only provides lower sea spray mass

retrievals because UHSAS-only fits are constrained solely by the accumulation mode shoulder. Sea spray mass concentrations from both methods exhibit generally consistent concentrations during the clean marine periods of LASIC with no apparent seasonality (Fig. 7a), but there are distinct differences in the retrieved mode diameters and widths (Fig. 7b,c).

Sea spray mode retrievals using UHSAS-only were much smaller in mean mass diameter than UHSAS-NEPH with averages

of $0.68 \pm 0.01$ µm and $1.3 \pm 0.15$ µm respectively (Fig. 7b). The range of mass mean diameters were similar for both retrieval methods, 0.4 to 1.6 µm for UHSAS-only and 0.6 to 1.6 µm for UHSAS-NEPH, though the majority of UHSAS-only mean diameters were submicron. Just over 1% (14) of UHSAS-only retrievals had mass mean diameters in the coarse mode (> 1 µm) compared to 92% (847) of UHSAS-NEPH. The peak in mass mean diameters at sizes within the coarse mode using UHSAS-NEPH can be attributed to the additional contributions of supermicron mass identified by the nephelometer

supermicron scattering that are not constrained by the UHSAS submicron size distribution. These predominately supermicron mean diameters are consistent with the assumption of sea salt particles contributing a large amount of mass and scattering within the coarse mode, which is simply not captured by the submicron UHSAS distributions alone.

On average, UHSAS-only retrievals were narrower at $1.8 \pm 0.4$ than those retrieved with UHSAS-NEPH at $2.2 \pm 0.2$ (Fig.

7c). The range of mode widths using UHSAS-only varied from particularly narrow (1.3) to a fairly broad and unconstrained width of 5.3, compared to 1.2 to 3.7 for UHSAS-NEPH. UHSAS-only widths are determined solely by the shape of the large accumulation mode shoulder in the UHSAS number size distribution and include a variety of widths based on how flat or sharp the slope of this shoulder may be. Mode widths retrieved from both methods were predominately narrower than a value of 3 with only 2% of UHSAS-only modes and 1% of UHSAS-NEPH modes greater than this value. Average UHSAS-only mode

narrowness again reflects the absence of supermicron size distribution measurements. (Sanchez et al., 2021) reported a similarly narrow average mode width ($1.44 \pm 0.25$) for marine boundary layer sea spray aerosol retrieved with UHSAS-only.



Conversely, (Yu et al., 2019) reported a broad mode width (geometric standard deviation of 2.7) and volume peak at supermicron diameter (approximately 2 µm) for 24-hr average sea salt particle size distributions at Ascension Island using Aerosol Robotic Network retrievals, which is more consistent with UHSAS-NEPH observations. These results indicate that

UHSAS-only may provide a good estimate of sea spray number concentration, which predominantly consist of submicron-sized particles (Sanchez et al., 2021), but the lack of supermicron measurements makes it unable to adequately identify mass contributions from larger particles.

Comparing the sea spray mode fitting parameters to those found in literature shows that retrieved modal properties for both

methods are within the range of reported values of mass mean diameter (0.25 – 1.6 µm) and mode width (1.4 – 3) (Fig. 7b,c and Table 5). The median of reported mode mass diameter (0.88 µm) falls in between the statistical mode values (peaks in histograms) of the retrieval methods at the upper end of UHSAS-only and lower end of UHSAS-NEPH (Fig. 7b), showing consistency with other sea spray mode measurements. Retrievals using only the UHSAS number size distributions generally show better agreement in terms of mode size with laboratory-based bubble bursting and breaking wave flume studies, which

had mass mean diameters that were less than 1 µm. UHSAS-NEPH retrievals are more consistent with field observations of the sea spray size distribution across several ocean basins, including open ocean studies in the Pacific and Atlantic in which measurements of supermicron size distributions were incorporated. The UHSAS-NEPH retrieval of sea spray mode width was on the lower end of the reported laboratory and field measurement values, while the narrow UHSAS-only modes were generally outside the majority spread. These differences and ranges in retrieved values show that the mode width is the least constrained

parameter derived by UHSAS-only and UHSAS-NEPH, although the scattering-based constraint provides some apparent improvement compared to UHSAS-only when supermicron mass contributions are considered.

Figure 8 displays six UHSAS-NEPH sea spray mode fits that are characteristic of the retrieval procedure applied to clean marine background observations during LASIC. Cases of different modal properties (diameter, width, mass) and the

comparison of this retrieval with the UHSAS-only algorithm are presented. For both methods, the accumulation mode (0.4 – 1 µm) is generally well characterized by the retrieved modes following the shape of this broad shoulder closely. Differences between the UHSAS-only and UHSAS-NEPH retrieved mass size distributions become more apparent just before the 1 µm size limit of the UHSAS. For narrow modes ($\sigma_g < 2$), using the UHSAS-only method appears to generally be sufficient for quantifying sea spray mode mass concentration in the absence of supermicron scattering measurements (Fig. 8a,b). In these

cases, the low contribution of coarse particles measured by the nephelometer supermicron scattering adds little information at the tail of the size distribution. The limitations of fitting a sea spray mode based solely on the shape of the accumulation mode "shoulder" in the number size distribution are illustrated in Fig. 8c,d,f. The broadness of the shoulder at submicron sizes (0.3 – 0.8 µm) forces the UHSAS-only retrieval to include more particles from submicron sizes and fewer from the supermicron regime. This leads to lower mass concentrations in UHSAS-only compared to UHSAS-NEPH. Mode retrievals using only the

UHSAS size distribution likely underestimate much of the mass at supermicron sizes as seen in the UHSAS-only modes tailing





off more sharply in the coarse regime (Fig. 8,c,f), with up to 5 µg m$^{-3}$ of sea spray mass being underestimated in the cases considered here.

**5.2 Sea Spray Tracers**

We next compared sea spray mode mass concentrations from UHSAS-only and UHSAS-NEPH with available sea spray tracers
to evaluate the extent to which the retrieved modal properties represent realistic sea spray size distributions.

The submicron mass concentration measured by the ACSM provided a trace chloride signal that could be used to examine sea salt mass concentrations from the UHSAS-only and UHSAS-NEPH retrievals. Comparing retrieved submicron sea spray mass with the ACSM chloride mass, we found correlations of R = 0.43 (p < 0.05) for UHSAS-only and R = 0.32 (p < 0.05) for
UHSAS-NEPH (Fig. 9a,b). While the ACSM signal is a very indirect chemical measurement of refractory chloride, these positive correlations provide additional support for the ability of both methods to identify retrieved modes as sea spray. The higher correlation of UHSAS-only over UHSAS-NEPH may be a result of the submicron sampling range of the ACSM, the diameter range in which UHSAS-only is solely constrained. Additionally, much of the sea spray mass retrieved from UHSAS-NEPH is concentrated in the supermicron regime, which is not observed in ACSM submicron chloride measurements.


Correlations between sea spray mass and the supermicron scattering coefficient were assessed for UHSAS-only and UHSAS-NEPH using retrievals that were deemed successful following the criteria discussed in previous sections. Sea spray mass concentrations from UHSAS-only were consistently less than 2 µg m$^{-3}$ and were not able to quantitatively explain the variability in the measured supermicron scattering at 550 nm, indicated by a very weak coefficient of determination (R$^2$ = 0.02,
p < 0.05; Fig. 9c). Using the number-size fitting algorithm, (Saliba et al., 2019) demonstrated a stronger correlation (R$^2$ = 0.63) between supermicron scattering and sea spray mass for ship-based measurements of the North Atlantic Aerosols and Marine Ecosystems Study than what is reported here for UHSAS-only during LASIC. However, in that study the inclusion of measured supermicron size distributions in the fitting procedure up to 10 µm diameter (compared to the submicron-only limit of the UHSAS) allowed for a better constraint on the retrieved supermicron sea spray particle mass. Although the stronger
correlation between sea spray mass and scattering with UHSAS-NEPH (R$^2$ = 0.72, p < 0.05, Fig. 9d) is an expected result given that the retrieved modes are constrained by scattering measurements, the lack of correlation from UHSAS-only further emphasizes the importance of including supermicron particle size measurements to adequately characterize how optical properties are influenced by the sea spray size distribution.

The functional relationship between sea spray mass concentration and wind speed was evaluated using linear regression on UHSAS-only and UHSAS-NEPH retrievals and resulted in correlation coefficients of 0.1 and 0.2, respectively (p < 0.05) (Fig. 9e,f). These correlations were lower than values in the range of 0.4 to 0.9 reported for numerous basins of the global ocean (Liu et al., 2021; Russell et al., 2010; Feng et al., 2017; Saliba et al., 2019). The calm and generally invariant wind speed





observed during the LASIC background season (7.1 ± 1.4 m/s) could explain the poor correlations for both methods as the
lack of dynamic range in wind speed at Ascension Island reduces the degree to which it explains the variability in concentration.
The variability may instead be impacted by marine airmass transport of sea spray aerosol from source regions away from
Ascension that reach the island rather than local wind conditions, which would not be resolved in the correlation (Grythe et
al., 2014). The improved correlation of UHSAS-NEPH in comparison to UHSAS-only shows some added value of applying
the supermicron scattering constraint to characterize the sea spray mode. Extending this method to a case study in the North
Atlantic where more dynamic wind conditions are observed shows that use of the supermicron scattering constraint is
comparable to the constraint of measured supermicron size distributions when estimating sea spray production from wind
speed (Text S2, Fig. S6g) as has been done in previous work (Saliba et al., 2019; Modini et al., 2015; Quinn et al., 2017).
These results indicate that the incorporation of nephelometer scattering as a constraint for supermicron particle size is provides
a reasonable replacement for measured supermicron mass size distributions.


## 6 Concluding Remarks

In this work we have presented a new method that combines measured submicron size distributions and 3-wavelength
supermicron scattering to estimate observationally-constrained sea spray modal properties at a remote marine site using a Mie
inversion (UHSAS-NEPH). When the retrieval was limited to clean marine periods ($CN_3 < 400$ cm$^{-3}$) reasonable sea spray size
distributions were obtained 95% of the time. UHSAS-NEPH had larger fit residuals for higher ambient scattering variability
($\Delta\sigma_{sca,RGB} > 5$ Mm$^{-1}$) and high relative humidity within the nephelometer (> 60%), which affected the consistency with which
the Mie solutions could be constrained by scattering measurements and (dried) UHSAS submicron mass size distributions.

Retrieved sea spray modes ranged in mean mass diameter from 0.6 to 1.6 µm (1.3 ± 0.2 µm), modal width from 1.2 to 3.7 (2.2
± 0.2), and mass concentration from 0.13 to 20.7 µg m$^{-3}$ (6.5 ± 3.5 µg m$^{-3}$), which are consistent with other field-based
measurements of the sea spray aerosol mode. By comparing retrieved modes to available tracers of sea salt, we have shown
that estimates of supermicron size, such as from Mie Inversion techniques, are necessary to resolve expected sea spray mass
correlations with chloride, scattering, and wind speed at Ascension Island. The observed positive correlation of UHSAS-NEPH
submicron sea spray mode mass with measured submicron chloride signal (R = 0.32) provided indirect chemical support of
UHSAS-NEPH modes as sea spray. UHSAS-NEPH showed stronger correlations ($R^2 = 0.72$) to the supermicron scattering in
comparison to fitting only based on the submicron size distribution accumulation mode ($R^2 = 0.02$), as expected from using
the scattering measurement to constrain UHSAS-NEPH solutions. Incorporation of scattering measurements as an estimate of
supermicron size improved the weak wind speed correlation (R = 0.2) relative to using only the submicron size distribution (R
= 0.1) for sea spray retrieval. This result was consistent with sea spray production even though the relationship was not as
strong as that found in prior studies with a greater dynamic range of wind speed and available supermicron size distributions.

Other environmental parameters such as sea surface temperature and its impacts on surface tension and kinematic viscosity could also be considered when assessing the relationship between sea spray production and modal properties relative to these retrieval methods at Ascension Island (Saliba et al., 2019; Liu et al., 2021; Salter et al., 2014), although NaCl chemical measurements would provide a more direct evaluation.


We have demonstrated that 3-wavelength scattering measurements constrained with submicron size distributions yield sea spray mode estimates that are consistent with sea salt during clean marine periods of LASIC. Inclusion of additional scattering wavelengths and chemical measurements of particles would provide additional constraints on refractive index and scattering efficiency by allowing for temporally resolved adjustments in parameters used for Mie simulations. The retrieval procedure

outlined in this work is a self-contained code with look-up table and is available to the broader scientific community. Future use of this combined size distribution and scattering-based approach to other measured marine datasets that lack supermicron size distributions can expand the array of sea spray observations and improve upon size, mass, and emission characterization of marine aerosol in climate models, further constraining the uncertainty in natural aerosol impacts on radiative forcing.

**Code Availability**

The sea spray mode retrieval algorithm and Mie theory scattering look-up table are available as a MATLAB function and MATLAB matrix file at the UCSD digital archives (https://doi.org/10.6075/J0GT5NCR). The Mie codes used to simulate sea spray scattering are available at the same location.

**Data Availability**

All LASIC data are publicly available from the ARM data discovery (https://adc.arm.gov/discovery/) (last access: 5 January

2022). Specific direction to each measurement dataset are provided as DOI references in Table 1. Retrieved sea spray modal parameters using LASIC and NAAMES measurements can be found at the UCSD digital archives (https://doi.org/10.6075/J0GT5NCR).

**Author Contribution**

Conceptualization and Methodology: **JLD, GS, LMR**. Code development, Formal Analysis**: JLD, GS, ASW**. Supervision

and Funding acquisition: **LMR and DL**. Writing – original draft: **JLD**. **All authors** contributed to the review and editing of the manuscript.

**Competing interests**

The authors declare that they have no conflict of interest.



**Financial Support**

This research was supported by the Director, Office of Science, Office of Biological and Environmental Research, Climate and Environmental Sciences Division of the U.S. Department of Energy under Contract DE-SC0021045.

**Acknowledgements**

The authors graciously thank Paquita Zuidema, the LASIC science team, and ARM instrument mentors for their assistance in the interpretation of measured size distributions from SMPS and UHSAS, and nephelometer scattering measurements. The
authors express thanks to Ian Eisenman for sharing computing resources that made the execution of Mie scattering simulations and sea spray retrieval possible. We also extend our thanks to Roya Bahreini for productive discussions on ways to incorporate measurement uncertainties and variabilities into the Mie Inversion methodology.

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



**Table 1.** LASIC measurements analyzed in this study.

| Variable | instrument | measurement | temporal resolution | availability[1] | data access |
|---|---|---|---|---|---|
| Particle Size Distributions | | | | | |
| | Ultra-High Sensitivity Aerosol Spectrometer (UHSAS) | dry particle size distributions $D_p$: 60 – 1 µm (optical diameter) | 1 Hz | 2016 Nov. – 2017 May | https://doi.org/10.5439/1333828 |
| | Scanning Mobility Particle Sizer (SMPS) | dry particle size distributions Dp: 0.01 – 0.46 µm (mobility diameter) | 5 min | 2016 Nov – 2017 May | https://doi.org/10.5439/1225453 |
| Particle Scattering | | | | | |
| | TSI 3563 3-wavelength Integrating Nephelometer (NEPH) | < 1 µm and < 10 µm dry total scattering coefficients at 450, 550, and 700 nm wavelengths | 1 min (alternating impactor cutoff approximately every 55 min) | 2016 Nov. – 2017 May | https://doi.org/10.5439/1259232 |
| Ancillary | | | | | |
| | Aerodyne Research Aerosol Chemical Speciation Monitor (ACSM) | mass concentration of non-refractory submicron chloride aerosol | 15 min | 2017 Jan. – 2017 May | https://doi.org/10.5439/1762267 |
| | TSI Ultrafine Condensation Particle Counter 3776 | condensation nuclei concentration of particle > 3 nm | 1 min | 2016 Nov. – 2017 May | https://doi.org/10.5439/1046186 |





| Vaisala WXT-520 | wind speed and rain intensity | 1 min | 2016 Nov. – 2017 May | https://doi.org/10.5439/1025153 |
| --- | --- | --- | --- | --- |
| DMT Single Particle Soot Photometer | refractory black carbon concentration | 15 min | 2016 Nov. – 2017 May | https://iop.archive.arm.gov/arm-iop/2016/asi/lasic/sedlacek-sp2/ |
| Los Gatos Research trace gas analyzer | carbon monoxide mixing ratio | 1 min | 2016 Nov. – 2017 May | https://doi.org/10.5439/1046183 |
| Radiance Research Particle Soot Absorption Photometer (PSAP) | < 1 µm and < 10 µm dry total absorption coefficients at 470, 552, and 660 nm wavelengths | 1 min | 2016 Nov. – 2017 May | https://doi.org/10.5439/1339528 |

[1]availability defined during typical Ascension Island background season (November – May).





**Table 2.** Scattering and size distribution measurements and the associated uncertainties and variabilities used in the sea spray retrieval method.

| | Measurement | | Variability or Uncertainty | |
|---|---|---|---|---|
| Scattering | | | | |
| | $b_{sca,1\text{-}10\mu m}(\lambda)$ | supermicron scattering at red (700 nm), green (550 nm), and blue (450 nm) light wavelengths | $\sigma_{sca,1\text{-}10\mu m}(\lambda)$ | standard deviation of supermicron scattering for averaging time at each wavelength |
| | | | $\sigma_{sca,inst}$ | instrument-defined scattering uncertainty (5%) |
| Size Distribution | PNSD | particle number size distribution | $\sigma_{PNSD,meas}(D_p)$ | standard deviation of PNSD resolved at each size bin |
| | | | $\sigma_{PNSD,inst}$ | instrument-defined concentration uncertainty (10%) |
| | $D_p$ | PNSD diameters | $\sigma_D$ | instrument-defined sizing uncertainty (2.5%) |






**Table 3.** Lognormal mode fitting parameters and resolution (step) used to derive Mie scattering sea spray mode solutions.

| Parameter | unit | minimum value | maximum value | step |
|:---:|:---:|:---:|:---:|:---:|
| $N_t$ | cm$^{-3}$ | 1.0 | 99 | 2.0 |
| $D_g$ | µm | 0.05 | 1.19 | 0.015 |
| $\sigma_g$ | | 1.0 | 4.0 | 0.075 |





**Table 4.** UHSAS-only and UHSAS-NEPH number concentrations, mass concentrations, and size distribution fitting
parameters. Values are mean ± 1 standard deviation. Bracketed values are the minimum and maximum ranges.

|  | UHSAS-only | UHSAS-NEPH |
| --- | --- | --- |
| $N_T$ (cm$^{-3}$) | 8 ± 7 | 6 ± 4 |
|  | [0, 151] | [0, 47] |
| $M_T$ (µg m$^{-3}$) | 8 ± 7 | 6.6 ± 3.5 |
|  | [0.008, 42] | [0.13, 20.7] |
| $D_{g,number}$ (µm) | 0.42 ± 0.10 | 0.49 ± 0.10 |
|  | [0.050, 0.54] | [0.10, 0.89] |
| $D_{g,mass}$ (µm) | 0.68 ± 0.01 | 1.3 ± 0.15 |
|  | [0.4, 1.6] | [0.6, 1.6] |
| $\sigma_g$ | 1.8 ± 0.4 | 2.2 ± 0.2 |
|  | [1.3, 5.3] | [1.2, 3.7] |





**Table 5**. Literature reported values of sea spray modal parameters. Number mean diameters ($D_{g,number}$) were converted to mass mean diameters ($D_{g,mass}$) using Eq. (7), integrating over particle sizes 0.01 – 10 µm, and averaging over a total particle concentration range of 1 – 100 cm$^{-3}$. Values are averages unless noted as an upper or lower bound.

| | | | Parameter | | |
|---|---|---|---|---|---|
| **Reference** | **Experiment type** | **Ocean basin** | **$D_{g,number}$ (µm)** | **$D_{g,mass}$ (µm)** | **$\sigma_g$** |
| Lewis and Schwartz (2004) | field measurements | | 0.3 | 1.3 | 3 |
| Sellegri et al. (2006), Keene et al. (2007), Fuentes et al. (2010), Modini et al. (2010), Bates et al. (2012), Zabori et al., (2012) | laboratory-based bubble bursting | | 0.05 (lower bound) 0.1 (upper bound) | 0.25 (lower bound) 0.48 (upper bound) | 2.8 |
| Prather et al. (2013) | laboratory-based breaking wave flume | N.E. Pacific | 0.16 | 0.88 | 3 |
| Modini et al. (2015) | field measurements | N.E. Pacific | 0.14 (lower bound) 0.26 (upper bound) | 0.5 (lower bound) 1.3 (upper bound) | 2.5 (lower bound) 3 (upper bound) |
| Quinn et al. (2017) | field measurements | Pacific, Southern, Arctic, and Atlantic | 0.3 | 1.08 | 2.5 |
| Saliba et al. (2019) | field measurements | N. Atlantic | 0.5 | 1.6 | 2.3 |
| Sanchez et al. (2021) | field measurements | Southern Ocean | 0.6 | 0.71 | 1.4 |





| This study | field measurements | S. Atlantic | 0.4 (UHSAS-only) 0.5 (UHSAS-NEPH) | 0.68 (UHSAS-only) 1.3 (UHSAS-NEPH) | 1.8 (UHSAS-only) 2.2 (UHSAS-NEPH) |





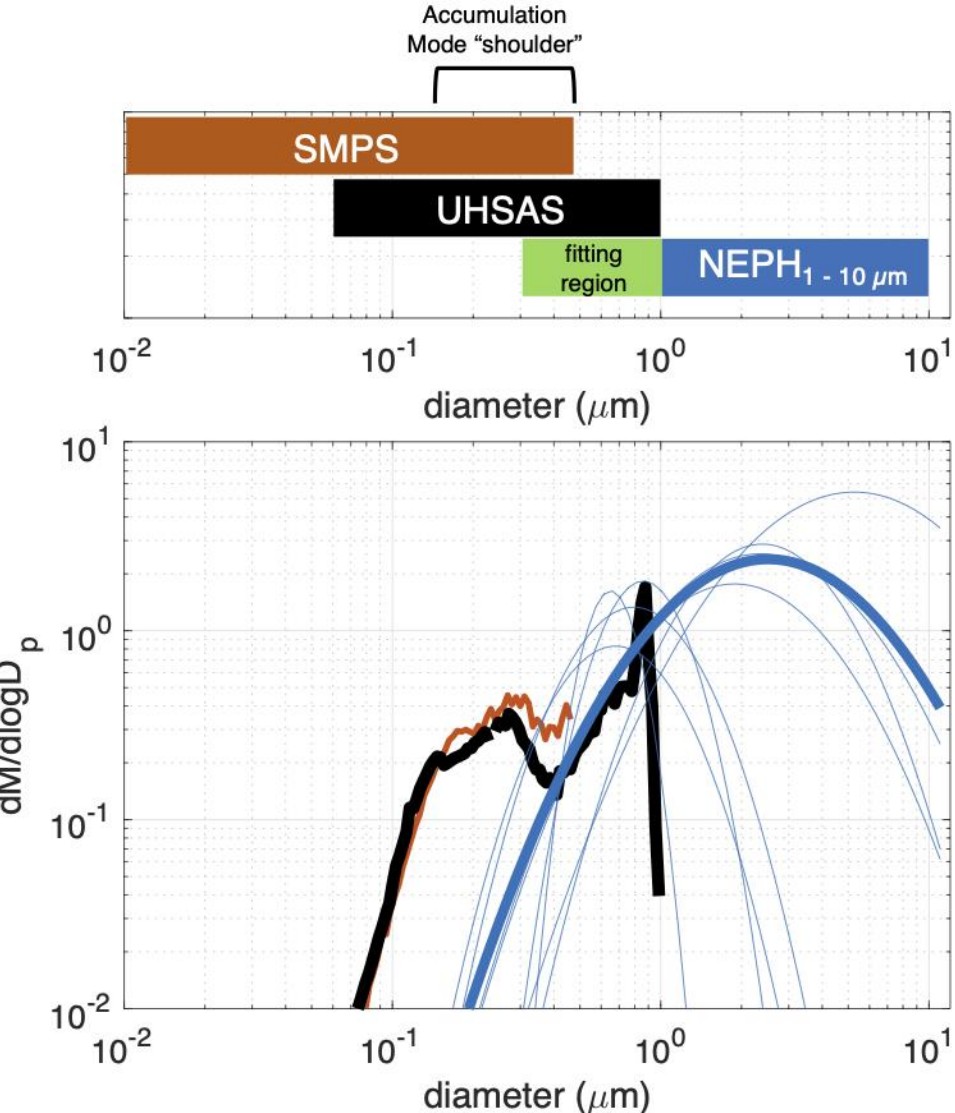


**Figure 1**. Schematic demonstrating the sea spray mode retrieval method using Mie theory-simulated size distributions, 3-wavelength integrating nephelometer supermicron scattering measurements, and UHSAS submicron mass size distributions (UHSAS-NEPH). The retrieval shown is for a 2-hr averaging period beginning 29 November 2016 14:00 UTC. (top panel) Instrument size ranges and mode fitting region for size distributions. (bottom panel) Mass size distributions ($\mu g\ m^{-3}$) measured by the SMPS (orange) and UHSAS (black), probable Mie theory-simulated lognormal sea spray mode solutions (thin blue), and best constrained Mie solution (thick blue). Note the UHSAS instrument artifact at $D_p = 0.85\ \mu m$ (see text for description).





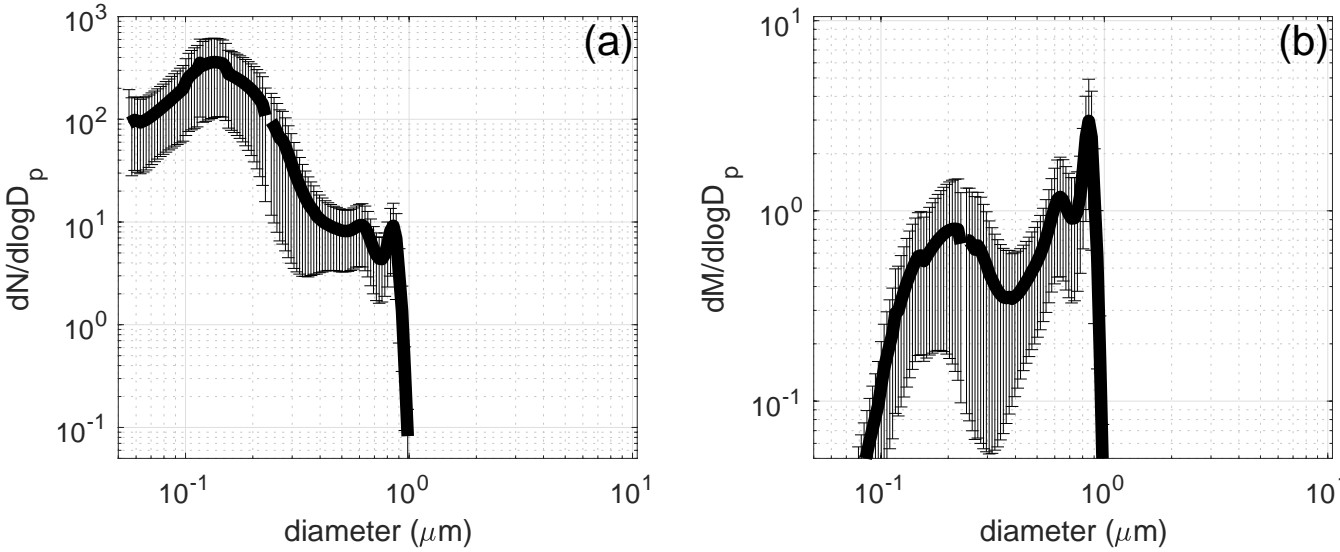

**Figure 2.** Average (solid black line) and variability (1 standard deviation; error bars) of the UHSAS (a) number (cm$^{-3}$) and (b) mass (µg m$^{-3}$) size distributions during the clean marine background season of LASIC (November 2016 – May 2017).



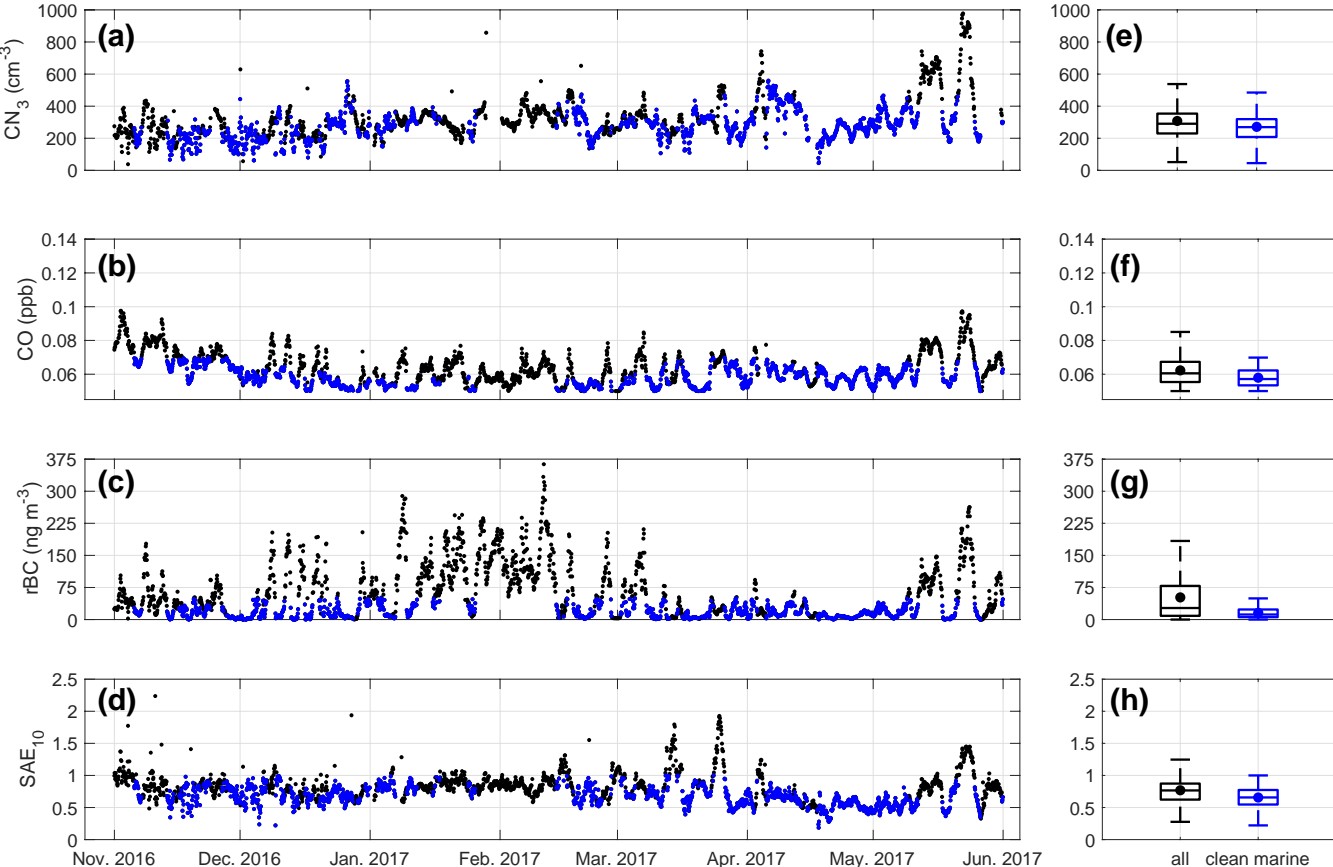

**Figure 3.** Time series (a-d) and box and whisker plots (e-h) of 2-hr average variables used to determine clean marine periods during the LASIC background season (November 2016 – May 2017). Periods that meet the criteria thresholds described in Section 2.2 are symbolized by blue dots. Circles within the box and whisker plots are the means and horizontal lines are the median and interquartile ranges (25% and 75%) for the background season (black) and clean marine periods (blue).





Figure 4. Flow chart describing the UHSAS-NEPH retrieval algorithm. Direction to descriptive procedures of each step are identified by main text section at right (below) the boxes.


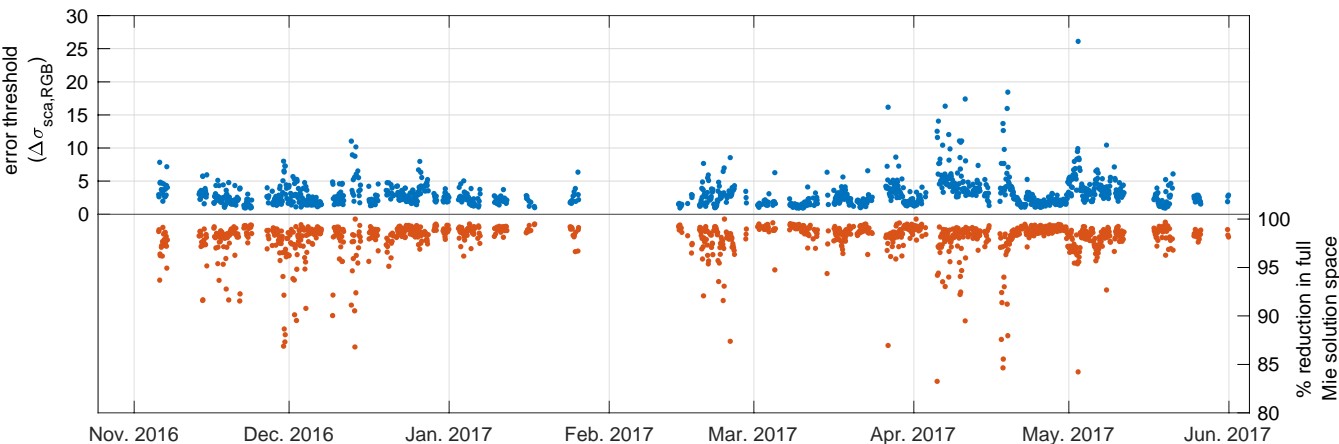

**Figure 5**. (top, blue) Time series of the scattering error threshold ($\Delta\sigma_{sca,RGB}$, Mm$^{-1}$) and (bottom, orange) percent reduction of the Mie look-up table solution space (N = 157,850) for UHSAS-NEPH retrievals during the background season of LASIC.






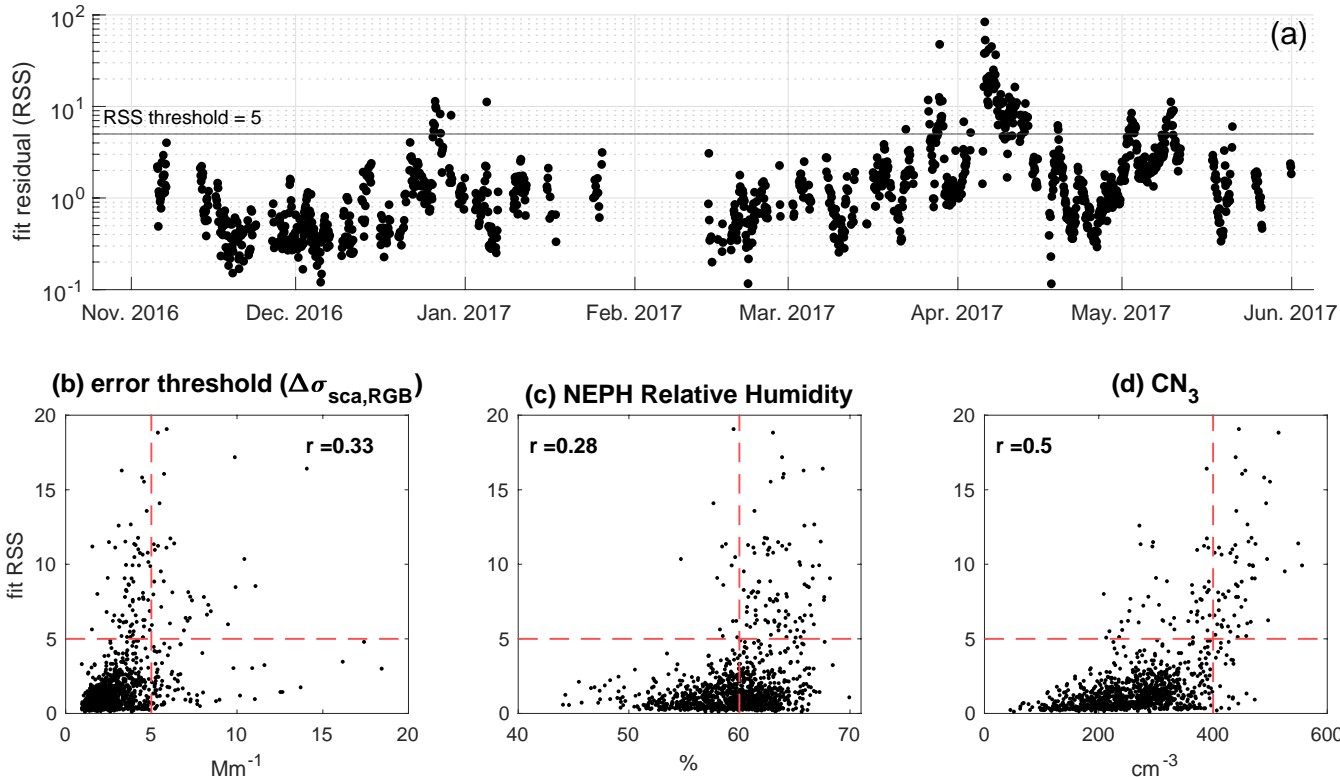

**Figure 6**. (a) Times series of the fit residual (residual sum of squares, RSS) between the retrieved sea spray mode and UHSAS mass size distribution within the $0.38 - 0.83$ µm fitting region. Horizontal line delineates RSS restriction threshold of 5 used in this procedure. Note logarithmic y-axis. (b-d) Selected variables used to restrict sea spray mode retrievals defined by the fit residual (fit RSS, y-axes). (b) error threshold ($\Delta\sigma_{sca,RGB}$, Mm$^{-1}$), (c) relative humidity within the nephelometer (%), and (d) condensation nuclei concentration of particles > 3 nm optical diameter (cm$^{-3}$). Retrieval restriction thresholds are symbolized by dashed red lines. Pearson correlation coefficients ($p < 0.05$) are provided within each panel.





**Figure 7.** (a) Time series of sea spray mass concentration (M_SS, µg m⁻³) retrieved from UHSAS-NEPH (blue) and UHSAS-only (orange) during clean marine periods of the LASIC background season (see Table 4 for summary statistics). Histograms of the (b) geometric mean mass diameter ($D_g$) and (c) geometric standard deviation ($\sigma_g$ mode width) for the retrieved sea spray modes using UHSAS-NEPH and UHSAS-only. Box and whisker plots above the histograms represent the quartile ranges (25th and 75th) and median (vertical) of sea spray mode fitting parameters reported in laboratory and field measurements (see Table 5).





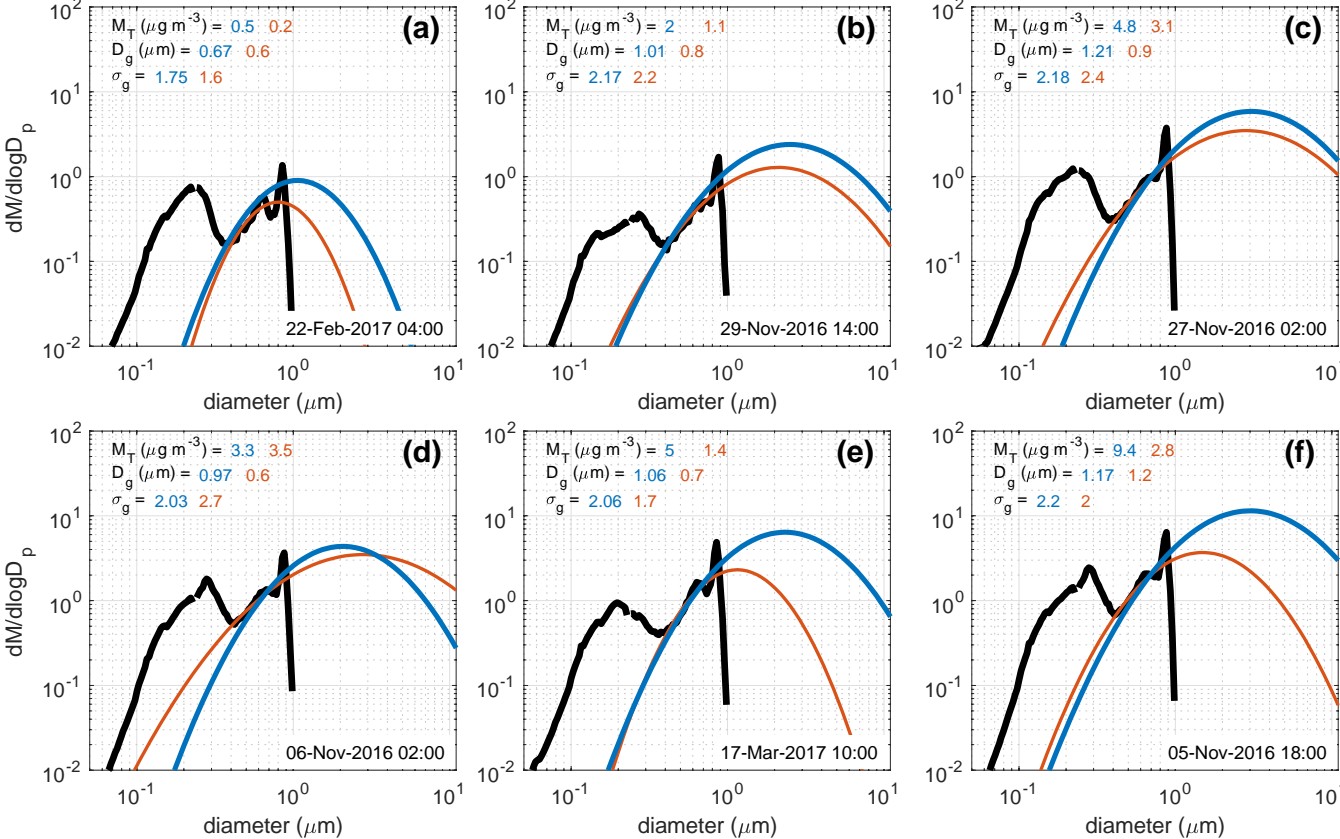


**Figure 8.** Selected characteristic fits to the sea spray mode. UHSAS mass size distribution (black), UHSAS-NEPH (blue), and UHSAS-only (orange). Panels are oriented to reflect increasing mass, size, and width of the sea spray mode retrieved by UHSAS-NEPH from left to right (a-c, d-f). Mode fitting parameters for the UHSAS-NEPH and UHSAS-only methods are identified in text and specified by method color within each panel: total mode mass ($M_T$), geometric mean mass diameter ($D_g$),
and geometric standard deviation ($\sigma_g$). 2-hr average time stamps are provided at the lower right.





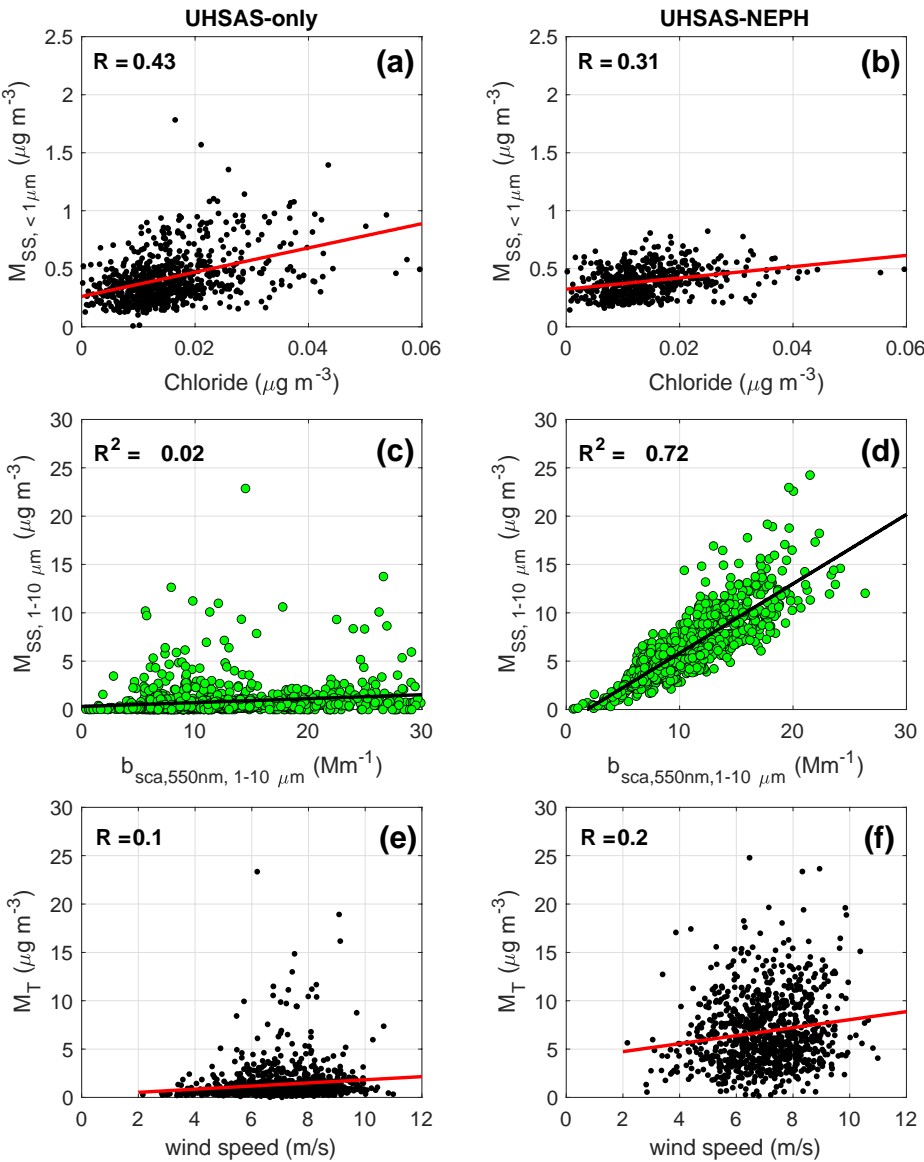

**Figure 9**. Correlations of submicron sea spray mass ($M_{SS,<1\mu m}$; μg m$^{-3}$) with ACSM chloride (a,b), supermicron sea spray mass
($M_{SS,1-10\mu m}$; μg m$^{-3}$) and measured supermicron scattering at 550 nm (c,d), total sea spray mode mass ($M_T$, μg m$^{-3}$) with wind
speed (e,f) for UHSAS-only (left) and UHSAS-NEPH (right) methods. Linear regressions are symbolized by black (red) lines
and coefficients of determination (correlation coefficients) are provided inside of the panels.