# Peer review of "Retrieval of the Sea Spray Aerosol Mode from Submicron Particle Size Distributions and Supermicron Scattering during LASIC"

_Atmospheric Measurement Techniques, 2022_

## Referee Comment (RC2)

Review of "Retrieval of the Sea Spray Aerosol Mode from Submicron Particle Size Distributions and Supermicron Scattering during LASIC", by J. Dedrick, et al., submitted to Atmospheric Chemistry and Physics.

This study used the particle number size distribution from UHSAS and scattering coefficients at 3 wavelengths from Nephelometer to construct the size distribution of sea spray aerosols. This topic is of great scientific interest. However, I have some concerns about the method, evaluation, and presentation of this study, before it can be considered for publication.

Major:
1. The instruments in LASIC, i.e. Neph, UHSAS, PSAP, are not performed under dry conditions, while at different RHs. I have three main concerns regarding the comparability of the different parameters either measured or calculated in this study:
    1) The mean RH of Neph is ~60% while UHSAS is ~55%, so this method is actually constraining the Mie simulated scattering with the measurements under different RHs. How will the RH difference influence the results? The scattering coefficient can be quite different for aerosols at 55% and 60% RH due to hygroscopic growth, especially for the strongly hydrophilic sea spray aerosols. The 5% RH can not be simply ignored just because "the uncertainty of the RH values (~10%) likely falls within the mean and range of reported humidity for each instrument" (page 5 line 150 to 154). Since this is the basis of this approach, more discussions and evaluations concerning the RH difference between Neph and UHSAS are needed to make this study robust and convincing.
    2) This study used 1.56+0i from previous studies for the refractive index, while the refractive index also changes with RH, and eventually influence the simulated scattering coefficients. Authors should take it into consideration in the Mie calculation.
    3) Authors compared SSA with those in previous studies; please make sure these values are comparable. The SSA in this study is not under dry conditions and might be overestimated as a result of the enhanced scattering coefficients due to aerosol hygroscopic growth.
    4) Section 4 authors stated "supermicron scattering was independent of the instrument RH up to a values of approximately 60%", while it showed an apparent increase of scattering coefficients with RH from 50% to 66% RH in Fig. S5.

2. In the evaluation section, we do not notice any significant improvements of the UHSAS-Neph method compared to the UHSAS-only method, except for the correlation between scattering coefficients with the sea spray particle mass (Fig. 9c and 9d). However, this correlation should be good for the UHSAS-Neph method as authors are using the scattering to constrain sea spray particle size distribution. Besides the good correlation with scattering, the UHSAS-Neph method is only slightly better (0.2 vs 0.1 for R) than the performance of the UHSAS-only method in the correlation with wind speed and worse in the correlation with the chloride mass

concentration of submicrometer particles. I would recommend moving S2 to the main context to help evaluate this method.

3. In-situ measurements found dust is one of the most observed aerosols at Ascension Island (Schenkels, 2018; Swap et al., 1996). I cannot tell how dust is differentiated from sea spray aerosols in this study. Please clarify.

4. The last paragraph of the introduction gives a detailed description of the method, which should be moved to the method part. The second paragraph of the introduction should be expanded to give a general review of previous studies. In particular, this study compared the results with those from the UHSAS-only method, which should definitely be introduced in the introduction.

5. I think this paper is missing a finalizing step. There are lots of grammar, citation, and reference format mistakes (e.g. line 49, line 434, line 470, line 472, and line 686). As there are lots of native speakers in the author list, this should not be a problem. I strongly suggest authors to do a thorough reading of this paper.

6. The authors used the range of "shoulder" to predict sea-spray number size distribution. However, this range is not consistent throughout the paper. In Fig. 1, it shows 0.1-0.4 μm; while it changed to 0.4-1 μm in line 124, and to 0.38-0.9 μm in line 389, and again to 0.38-0.83 μm in line 392. The authors need to explain the variation of the shoulder range.

7. Nucleation mode, Aitken mode, accumulation mode, and coarse mode are four modes used to describe the aerosol number size distribution. This paper used a lot of "sea spray mode" or "sea spray aerosol mode", which is not common usage, I suggest authors to revise.

8. In Section 3.4, the authors converted number size distribution to mass size distribution by assuming the density to be 1g cm$^{-3}$ for sea spray aerosols. This value seems to be quite low, less than half of the value of 2.017g cm$^{-3}$ used in Zieger et al. (2017). Please clarify the reasons for choosing this density.

Others:
1. Page 3 line 64, please clarify "relative availability".
2. Line 144, again, the Neph is not measured under dry conditions, I strongly suggest not to use "dry scattering coefficients".
3. Line 166, the citation "Dmt, 2017" is incorrect. Please revise according to the journal's citation and reference format.
4. I would suggest adding the ranges of $\sigma_{sca,1-10\mu m}$ and $\sigma_{PNSD,mea}$ in Table 2.
5. Please clarify "MAOS" in line 181.
6. I suggest authors to specify under which RHs different variables are measured in Table 1.
7. The threshold of 0.07 ppb for CO seems extremely low (line 229 and Fig. 3).
8. Please add references in line 271, 314, 519, and 522.
9. I would suggest moving S1 to the main context.

10. Authors use a lot of terms in MATLAB, e.g. normrn, I would suggest avoiding using them or adding explanations for these terms.
11. Please clarify the "low error-restricted Mie solution".
12. Please explain why "These high fit RSS likely indicate that the supermicron scattering measurements were influenced by particles other than sea spray" in line 397 and "had fit RSS values above 5, possibly because they may have been influenced by dust or other non-marine aerosol intrusions that were not effectively screened using the clean marine criteria" in line 404.
13. Line 400, please add a table in the supplement showing the variation of correlations with different thresholds.
14. Please clarify "no observations during this period" in line 419.
15. Please clarify the location of the description for the UHSAS-only method in line 469.
16. The value in line 483 is inconsistent with the one in Table 4.
17. In the evaluation section, the authors stated that the sigma from the UHSAS-only method is much broader than those from the UHSAS-Neph method; please clarify which sample from UHSAS-Neph is using here; the 1237 or the 971? If it is the 971 one, would the constraints in Section 4 help reduce the sigma range and result in the narrower range?
18. The discussion of Fig. 8 needs revision. In line 533, the authors stated $\sigma_g < 2$, while Fig. 8b is actually $>2$. No discussion for Fig. 8e?
19. In Fig. 9, the authors use $R^2$ for Fig. 9c and 9d, while R for others; why?

Schenkels, M.: Aerosol Optical Depth and Cloud Parameters from Ascension Island retrieved with a UV-depolarisation Lidar; An outlook on the validation, Master thesis, Utrecht University, 2018.

Swap, R., Garstang, M., Macko, S. A. et al: The long-range transport of southern African aerosols to the tropical South Atlantic, J. Geophys. Res. Atmospheres, 101, 23777–23791, https://doi.org/10.1029/95JD01049, 1996.

Zieger, P., Väisänen, O., Corbin, J. et al: Revising the hygroscopicity of inorganic sea salt particles. Nat Commun 8, 15883, https://doi.org/10.1038/ncomms15883, 2017.

---

## Author Response (AR1)

**Responses to Reviewer Comments**

We thank both reviewers for their interest in this work and for taking the time to provide detailed and constructive comments on our study. Several of the major comments raised by the reviewers have encouraged us to include more information on the sensitivities and uncertainties for sea spray size distribution retrieval. Although the major results and conclusions of this study were not significantly changed, these revisions have improved the manuscript and made the results more robust.

In the text below, reviewer comments are in black and our responses are in purple.

**Reviewer 1**

The paper presents a method of using the combination of submicron number size distribution and supermicron 3-wavelength nephelometer measurements to derive sea spray aerosol modal parameters. The measurements were made on Ascension Island with the analysis restricted to periods of clean marine conditions. Retrieved sea spray modal mass parameters (diameter, width, mass concentration) are compared to sea spray tracers (supermicron scattering, wind speed, and chloride) to assess the success of the method. It is found that the use of supermicron scattering improves the ability to derive sea spray modal properties above the use of only the submicron number size distribution. The paper is well-written and provides a way to retrieve sea spray mass size distribution properties when only a portion of the submicron number size distribution has been measured.

The most robust test of the method uses comparisons to measured supermicron size distributions from NAAMES 1. As such, Section Text S2, Figure S6, and Table S1 should be included in the main text and not relegated to the supplement material.

We thank the reviewer for noting the relevance of this section in supporting the use of this method. We agree that Supplemental Text S2 provides a valuable evaluation of the method and have moved this text to the main text as Section 4: Evaluation of NEPH-constrained Sea Spray Retrieval with Supermicron Size Measurements.

Lines 226 - 227: It is stated here that CN3 concentrations less than 600 were used to define clean marine periods. In the abstract, it is stated that particle concentrations less than 400 were considered typical of the clean marine boundary layer. Please clarify.

This sentence was not clearly stated in the abstract. The clean marine criteria, which includes the  $< 600 \text{ cm}^{-3}$  threshold, was applied to initially screen the data for only times representative of "clean marine" conditions. Comparing and correlating the fit RSS retrieval parameter and total particle concentration showed that a 400 cm-3 total particle threshold provided sea spray mode retrievals that had low fit residual to the measured size distribution and therefore provided an additional screening for the retrieved modal parameters. This passage of the abstract has been revised to more clearly state this:

The UHSAS-NEPH method retrieved sea spray mode properties for approximately 88% of background marine times when the scattering variability and total particle concentrations were low ( $< \pm 5 \text{ Mm}^{-1}$  and  $< 400 \text{ cm}^{-3}$ , respectively)

Lines 345 - 346: Is the statement that "the often strongly correlated relationship between supermicron scattering and sea spray mass" based on the results shown in Figure S1a or previously reported results? If the latter, please provide references.

This statement is based on previously reported results from literature. We have added the following references to this statement: (Chamaillard et al., 2006; Kleefeld et al., 2002; Quinn et al., 1998).

Lines 417 - 419: It appears that there is a word or two missing from this sentence. No observations during this period were used? Or no observations were eliminated?

This passage, which includes a discussion of dust screening (Lines 403 - 419), was based on the assertion that high fit RSS was associated with dust at the surface for the period of March 27 – April 4, 2017. The dust screening was applied only for this period and the assertion was not supported by the assessment of the single-scattering albedo and scattering angstrom exponents, which had values that were more consistent with sea spray rather than dust (SSA = 0.98, SAE = 0.46). Therefore we have removed these lines from the main text (see response to Reviewer 2 comments 12 and 14). Following the suggestion of Reviewer 2 (see major comment 3), we have chosen to apply the dust screening metric to all clean marine background observations so that periods of potential dust influences at the surface are removed. This screening removed 68 periods from the analysis and had no significant effects on the mean statistics of the retrievals. A description of the screening has been added as Supplemental Text 1 (see response to Reviewer 2 comment 3) and is briefly summarized in Section 2.2 as follows:

Saharan dust and continental aerosol transport from southern Central Africa into the remote tropical Atlantic boundary layer has been a commonly observed contributor to the surface-level aerosol population at Ascension Island (Swap et al., 1996). The mass concentration of transported dust particles are largely in the supermicron size range (Miller et al., 2021; Denjean et al., 2016) and overlap the fitting region used in UHSAS-NEPH. To exclude interference in the retrieval from dust particles, we used measurements of the sub-10 µm single-scattering albedo at 470 nm (SSA470nm) from NEPH scattering and PSAP absorption to identify times with possible dust influence (SI Text 1). An SSA470nm threshold of 0.95 was used to distinguish between sea salt and dust aerosol contributions to coarse scattering based on the relationship of SAE10 and average SSA reported for Saharan dust (Di Biagio et al., 2019; Von Hoyningen-Huene et al., 2009; Haywood et al., 2003). This restriction removed 68 2-h periods.

Lines 518 - 522: Please provide references for the laboratory, flume, and field observations that are compared to in these statements.

References have been added for these passages.

Line 583: change to "incorporation of nephelometer scattering as a constraint for supermicron particle size provides a reasonable"

This grammatical error has been corrected.

**Reviewer 2**

Review of "Retrieval of the Sea Spray Aerosol Mode from Submicron Particle Size Distributions and Supermicron Scattering during LASIC", by J. Dedrick, et al., submitted to Atmospheric Chemistry and Physics.

This study used the particle number size distribution from UHSAS and scattering coefficients at 3 wavelengths from Nephelometer to construct the size distribution of sea spray aerosols. This topic is of great scientific interest. However, I have some concerns about the method, evaluation, and presentation of this study, before it can be considered for publication.

Major:

- 1. The instruments in LASIC, i.e. Neph, UHSAS, PSAP, are not performed under dry conditions, while at different RHs. I have three main concerns regarding the comparability of the different parameters either measured or calculated in this study:
  - 1) The mean RH of Neph is ~60% while UHSAS is ~55%, so this method is actually constraining the Mie simulated scattering with the measurements under different RHs. How will the RH difference influence the results? The scattering coefficient can be quite different for aerosols at 55% and 60% RH due to hygroscopic growth, especially for the strongly hydrophilic sea spray aerosols. The 5% RH can not be simply ignored just because "the uncertainty of the RH values (~10%) likely falls within the mean and range of reported humidity for each instrument" (page 5 line 150 to 154). Since this is the basis of this approach, more discussions and evaluations concerning the RH difference between Neph and UHSAS are needed to make this study robust and convincing.

We thank the reviewer for noting the potential role of this difference between instrument relative humidity values and the effects this will have on the retrieval methodology in terms of particle size (hygroscopic growth) and simulated scattering properties (refractive index). In our restriction of nephelometer measurements to those with less than 60% RH, we failed to specify that our intention was to closely match the average and uncertainty of the NEPH to the UHSAS. Specifically the 60% upper cutoff yielded an average nephelometer relative humidity of 55%  $\pm$  2%. With the inclusion of the uncertainty in the reported RH, we further clarify this as 55  $\pm$  10%, meaning that the NEPH and UHSAS report comparable RH under this restriction. The UHSAS RH includes additional uncertainty since it is not a recorded measurement and was estimated from ambient and sample line measurements (a brief note on this has been added to the main text in Section 2.1.1 Submicron Particle Size Distributions).

Given this change in the text, it made sense to move the discussion of RH effects from Section 4 (Performance of UHSAS-NEPH Retrievals) to the discussion of the nephelometer measurements (Section 2.1.2 Supermicron Scattering). We also revised this analysis to remove periods that exceeded the 60% cutoff before performing UHSAS-NEPH retrievals. The 60% RH cutoff that was used originally and in this revised version resulted in the removal of 345 periods from the dataset, which reduced the total number of clean marine observations to 909.

The following clarification has been added to the main text (Section 2.1.2 Supermicron Scattering):

Particle scattering measurements during LASIC were not available at standard dry conditions (< 40%) as operating conditions only allowed for limited heating, typically producing 60  $\pm$  4% RH at the nephelometer inlet for ambient conditions of 88  $\pm$  8% RH. This average RH of 60% means particles were not dried to the efflorescence point for sea salt mixtures (< 40%) (Ming and Russell, 2001). The supermicron scattering (at 450 nm) did not show a significant correlation to the instrument RH (R = 0.22, p =0.19; Fig. S1\*), but the correlation increases to R = 0.36 (p < 0.05) for RH > 60 %, indicating that at higher RH the scattering may need to be corrected for humidity. Measurements of particle scattering at a series of pre-set RH were also collected during LASIC to provide hygroscopic growth factors (f(RH)) to correct scattering from 65% RH to the heated conditions (Zieger et al., 2010; Gasso et al., 2000). However, because the uncertainty for f(65% RH) was estimated to be > 30%, which was approximately fourfold greater than the 8% for  $< 1 \ \mu m$  and 7% for  $< 10 \ \mu m$  scattering uncertainties for the heated measurements, we did not apply this correction. This uncertainty was driven by the limited and non-overlapping times for which f(65% RH) was available for  $< 1 \ \mu m$  and  $< 10 \ \mu m$  scattering, each typically spanning only  $30 \ \pm$ 5% of the 2-h averaging period. Without sufficient and simultaneous f(RH) measurements of the humidity dependence of scattering, and given the additional uncertainties associated with correcting optical size distribution measurements with humidity and composition dependent refractive indices (Kassianov et al., 2015), correcting the measurements to a standard RH was also not possible. Instead, we restricted the measurements to include only those for which the average nephelometer humidity matched the average UHSAS RH. Restricting nephelometer measurements to those that had RH below 60% gives an average RH of 55% with 10% measurement uncertainty (55  $\pm$  10%), while still retaining 78% of the measurements for this analysis.

\*See Response to Reviewer 2 comment 4 for this figure.

Also, we note that our estimates of the sea spray mode in "humid" conditions are likely larger than those of dried sea spray mode properties. We additionally show in the following responses to comments by the reviewer that considerations of refractive index and particle density change within the RH range of the UHSAS and NEPH provided only marginal changes to the retrieved modal properties (increase in mass concentration and diameter), though using the assumed refractive index and particle density for an average RH of 55% provided some improvement to retrieved sea spray mass correlations with available tracers.

2) This study used 1.56+0i from previous studies for the refractive index, while the refractive index also changes with RH, and eventually influence the simulated scattering coefficients. Authors should take it into consideration in the Mie calculation.

The authors thank the reviewer for pointing out this concern. To address this comment, we evaluated the sensitivity of the mode retrieval to different refractive indices (*m*) that represent changes in the inlet relative humidity. *m* values were selected in the range of 1.4 + 0i to 1.6 + 0i. This range of *m* was based on laboratory and modelling experiments of *m* and Mie scattering of sea salt aerosol at varying relative humidity that ranged from < 40% (dry) to > 75% (wet) (Randles et al., 2004; Bi et al., 2018; Wang and Rood, 2008; Saliba et al., 2019). For the purposes of these tests, we assume the sea salt particles are largely inorganic mixtures with organic components contributing minor if not offsetting *m* differences. The imaginary component of *m* is neglected under the assumption that the clean marine criteria and single-scattering albedo restrictions limit influences of absorbing contributions from non-marine organics and mineral dust. We also assume that the wavelength dependence of *m* in the visible light range relevant to the nephelometer scattering (450 – 700 nm) is weak (Bi et al., 2018). The retrievals were also run after correcting the sea spray density as described in the response to Reviewer 2 major comment 8.

| Table 1. Refractive Index values used to define RH condition in Mie simulations. |                               |                        |                     |  |
|----------------------------------------------------------------------------------|-------------------------------|------------------------|---------------------|--|
| Refractive Index ( m )                                             | Relative Humidity (RH) | Reference              | remark              |  |
| 1.5 + 0i                                                                         | "dry"                         | (Bi et al., 2018)      |                     |  |
| $1.45 + 10^{-8}i$                                                                | 66%                           |                        |                     |  |
| $1.42 + 10^{-7}i$                                                                | 71%                           |                        |                     |  |
| $1.39 + 10^{-9}i$                                                                | 76%                           |                        |                     |  |
| 1.56 + 0i                                                                        | <40%                          | (Saliba et al., 2019)  | Assumed NaCl        |  |
| $1.51 + 10^{-7}i$                                                                | "dry"                         | (Randles et al., 2004) | Organic – inorganic |  |
|                                                                                  |                               |                        | mixture             |  |
| 1.54 + 0i                                                                        | "dry"                         | (Wang and Rood, 2008)  | NaCl                |  |

The results of these tests are provided in the following table:

| Table 2. Effects of refractive index changes on sea spray size distribution fit parameters and tracer correlations. |                                               |                             |               |                                                                       |                                                                           |                                                                    |                                        |
|---------------------------------------------------------------------------------------------------------------------|-----------------------------------------------|-----------------------------|---------------|-----------------------------------------------------------------------|---------------------------------------------------------------------------|--------------------------------------------------------------------|----------------------------------------|
| т                                                                                                                   | sea
spray
mass
(µg m -3 ) | D g,mass
(µm) | $\sigma_{g}$  | <pre>< 1 µm sea spray mass correlation (R) with < 1 µm Cl</pre> | < 10 µm sea
spray mass
correlation
(R) with <10
µm scattering | < 10 µm sea
spray mass
correlation
(R) with wind
speed | Average
fit
residual to
UHSAS |
| 1.4+0i                                                                                                              | 8.03 ±
3.65                                | 1.42 ±
0.16              | $2.3 \pm 0.2$ | 0.30                                                                  | 0.82                                                                      | 0.16                                                               | $1.25 \pm 0.83$                        |
| 1.45+0i                                                                                                             | 8.37 ± 4.07                                   | 1.47 ±
0.17              | 2.4 ± 0.3     | 0.35                                                                  | 0.84                                                                      | 0.20                                                               | 1.26 ±
0.89                         |
| 1.5+0i                                                                                                              | 8.89 ±
3.85                                | 1.47 ±
0.16              | 2.4 ± 0.2     | 0.34                                                                  | 0.85                                                                      | 0.20                                                               | $1.26 \pm 0.90$                        |
| 1.56+0i
(original)                                                                                               | 8.92 ±
3.30                                | $1.5 \pm 0.15$              | 2.4 ± 0.2     | 0.31                                                                  | 0.84                                                                      | 0.20                                                               | 1.26±
0.85                          |
| 1.6+0i                                                                                                              | 9.26 ±
4.19                                | 1.52 ± 0.17                 | 2.46 ± 0.3    | 0.31                                                                  | 0.83                                                                      | 0.17                                                               | 1.26±
0.83                          |

Differences between retrievals using different m were observed for the sea spray size distribution fit parameters, and mass correlations, although none of these results significantly altered the results found for the original retrieval using an m value of 1.56 + 0i. The retrieved sea spray mass concentrations using m that represent humidified

and wet conditions (1.4 + 0i, 1.45 + 0i) were lower on average than those using a "dry" m (1.5 + 0i, 1.6 + 0i), but they are within the range of variability for dry m. Retrieved sea spray mass correlations with submicron chloride and supermicron scattering were slightly improved when simulating Mie scattering with *m* assumed under more humid conditions (1.45 + 0i, 1.5 + 0i), potentially reflecting a more accurate sea spray retrieval when the simulated scattering is comparable in RH to the measured scattering (average  $RH = 55 \pm 10$  %) and UHSAS size distribution measurements ( $RH = 55 \pm 8$ %). The sea spray mass correlations with wind speed for all *m* were not improved beyond the original 0.2, but the most wet m(1.4 + 0i) and driest m(1.6 + 0i) cases saw reduction in correlation. The chloride and scattering sea spray tracer correlations showed improvement when simulating scattering with an m of 1.45 + 0i and 1.5 + 0i. Even though the changes between the retrieved fit parameters and tracer correlations do not vary substantially, we revised our Mie calculations to the *m* value of 1.45 + 0i to simulate the sea spray scattering and mode retrieval during LASIC, which more appropriately reflects the sea spray refractive index for salt particles at 55% RH. For the dry size distribution and scattering case using NAAMES data, the *m* value originally assumed for dry scattering and size distributions (1.56 + 0i) was used and provided reasonable retrieval of sea spray mode properties based on the evaluated correlations ( $

**Figure S1**. Scatter plot of nephelometer control humidity and supermicron scattering at 450 nm (Mm-1). The delineation of black and red points indicates the restriction used for applicable UHSAS-NEPH sea spray mode retrieval (Section 2). Linear (dashed blue line) and power law (solid blue line) fits have been applied to the data.

2. In the evaluation section, we do not notice any significant improvements of the UHSAS-Neph method compared to the UHSAS-only method, except for the correlation between scattering coefficients with the sea spray particle mass (Fig. 9c and 9d). However, this correlation should be good for the UHSAS-Neph method as authors are using the scattering to constrain sea spray particle size distribution. Besides the good correlation with scattering, the UHSAS-Neph method is only slightly better (0.2 vs 0.1 for R) than the performance of the UHSAS-only method in the correlation with wind speed and worse in the correlation with the chloride mass concentration of submicrometer particles. I would recommend moving S2 to the main context to help evaluate this method. We thank the reviewer for noting the relevance of this section in supporting the use of this method. We agree that Supplemental Text S2 provides a valuable evaluation of the method and have moved this text to the main text as Section 4: Evaluation of NEPH-constrained Sea Spray Retrieval with Supermicron Size Measurements.

3. In-situ measurements found dust is one of the most observed aerosols at Ascension Island (Schenkels, 2018; Swap et al., 1996). I cannot tell how dust is differentiated from sea spray aerosols in this study. Please clarify.

The references provided by the reviewer note frequent boundary layer dust intrusions at Ascension Island during austral spring and summer months (June – October), which typically coincide with the biomass burning season. In our initial screening of the data using the clean marine criteria, we chose to only focus on periods during the "background" season (nominally November – May), which is expected to have quiescent marine conditions and limited influence of dust at the surface. However, other studies have reported that boundary layer dust episodes can occur during the background season (January – April) (Kishcha et al., 2015; Virkkula et al., 2006). To preclude the possibility of dust during the marine background season, we have added a criterion to our data screening that removes potential dust influences using single-scattering albedo and scattering angstrom exponent thresholds, as reported in the section titled "Measurement Screening" (previously "Clean Marine Periods"). Further description of this method is provided in the Supplemental Text (below). Application of this dust screening removed only 68 of the 2-hr clean marine background LASIC observations and had no significant effects on the mean statistics of the retrievals.

Supplemental Text 1: Differentiating Sea Spray Aerosol from Saharan Dust

Absorbing particle influences from dust and other non-marine/continental aerosol were screened from the dataset using the sub-10  $\mu$ m single-scattering albedo (SSA). The SSA is defined as

$$SSA (\lambda) = \frac{b_{sca}(\lambda)}{b_{sca}(\lambda) + b_{abs}(\lambda)},$$
(S1),

where  $b_{sca}$  and  $b_{abs}$  are the scattering and absorption coefficients, respectively. Sea salt aerosol typically have a high SSA (~1), while the greater absorbance properties of dust and biomass burning decrease SSA to lower values (0.8 – 0.9) (Muller et al., 2011; Wu et al., 2020; Zuidema et al., 2018). Observational estimates of Saharan dust SSA provide a range of 0.91 – 0.97 for wavelengths between 400 and 550 nm (Di Biagio et al., 2019; Von Hoyningen-Huene et al., 2009; Haywood et al., 2003) and a mean across these wavelengths of 0.95. Due to the predominantly coarse sizes of sea salt and dust, both are expected to have low scattering angstrom exponent values (SAE < 1) (Delene and Ogren, 2002). We therefore distinguish between these two types of aerosol by identifying the regime in which both the SSA is high (~ 1) and the SAE is low (< 1) for sea salt dominated periods.

To estimate SSA, NEPH scattering was adjusted to match the PSAP wavelength of 470 nm using the sub-10  $\mu$ m SAE at 450-550 nm as described by De Faria et al. (2021). We note that the

absorption and scattering coefficients were measured at different relative humidity during LASIC (PSAP RH < 25%, Zuidema et al. 2018; NEPH RH = 55%). These differences may lead to overrepresentation of the SSA due to the larger NEPH humidity, but it provides a baselevel estimate of the parameter that can be used to distinguish the aerosol types. The SSA estimate was calculated for the full 2-h LASIC dataset (April 2016 – October 2017), which captures both the background (November – May) and biomass burning (June – October) seasons. The results relating the SSA and SAE are provided in Fig. S3. For non-marine periods (periods that did not meet criteria described in Sect. 2.2), the SAE tends toward larger values (> 1) and lower SSA (

**Figure S3**. Scattering plot of the 450-700 nm scattering Angstrom exponent at  $10 \,\mu\text{m}$  (SAE10) and the 470 nm single-scattering albedo (SSA470 nm). Blue circles represent measurements that meet the clean marine criteria identified in Section 2.2, while orange circles are non-marine times. The dashed vertical line marks an SAE10 value of 1 (criteria for clean marine screening) and the dashed horizontal line marks SSA470 nm 0.95 (criteria to distinguish sea salt from dust).

4. The last paragraph of the introduction gives a detailed description of the method, which should be moved to the method part. The second paragraph of the introduction should be expanded to give a general review of previous studies. In particular, this study compared the results with those from the UHSAS-only method, which should definitely be introduced in the introduction.

We thank the Reviewer for these suggestions. The last paragraph of the introduction has been revised to better introduce the paper topic. The second paragraph has been revised and expanded to give a more general review of previous work regarding the retrieval of sea spray size distributions such as the UHSAS-only method. The method details that were previously included in the last paragraph of the introduction have been moved to the methodology in Section 3.1 and Section 3.3.

5. I think this paper is missing a finalizing step. There are lots of grammar, citation, and reference format mistakes (e.g. line 49, line 434, line 470, line 472, and line 686). As there are lots of native speakers in the author list, this should not be a problem. I strongly suggest authors to do a thorough reading of this paper.

After addressing the major and minor comments raised by the reviewers, the paper has been carefully edited and reread to ensure that grammatical, citation, and reference format mistakes are removed.

6. The authors used the range of "shoulder" to predict sea-spray number size distribution. However, this range is not consistent throughout the paper. In Fig. 1, it shows 0.1-0.4 μm; while it changed to 0.4-1 μm in line 124, and to 0.38-0.9 μm in line 389, and again to 0.38-0.83 μm in line 392. The authors need to explain the variation of the shoulder range.

These variations in the shoulder range were due to a lack of proofreading and we thank the reviewer for catching the inconsistencies in the text and Figure 1. For consistency, the description of the "shoulder" region has been revised to  $D_p > 0.4 \,\mu\text{m}$ . For the specific case of amending this region in the UHSAS size distribution due to instrument artifacts, we have noted that the range is  $0.38 - 0.83 \,\mu\text{m}$ , (the closest UHSAS size distribution bins within the specified range).

We have revised this description in the text (Section 3.4) as follows:

To account for the consistent UHSAS artifacts at 0.6 and 0.85  $\mu$ m (Section 2.1.1), we restricted the measured size distributions used to fit the Mie theory-simulated size distributions for diameters larger than 0.4  $\mu$ m (0.4 to 1  $\mu$ m UHSAS range) to 0.38 – 0.83  $\mu$ m (the closest UHSAS diameter size bins within the specified range), which weights the comparison toward smaller sizes and effectively reduces the influence of the largest artifact while maintaining the shape of the accumulation mode "shoulder" (Fig. 1).

Figure 1 has been revised with corrections made to the position of the fitting region and accumulation mode shoulder.